# Adaptive Shrinkage Estimation for Personalized Deep Kernel Regression in Modeling Brain Trajectories

**Vasiliki Tassopoulou**[1,2]**, Haochang Shou**[1,3]**, and Christos Davatzikos**[1,2]
[1]Center for AI and Data Science for Integrated Diagnostics, University of Pennsylvania
[2]Department of Bioengineering, University of Pennsylvania
[3]Department of Biostatistics, Epidemiology and Informatics, University of Pennsylvania
{vasiliki.tassopoulou, hshou, christos.davatzikos}@pennmedicine.upenn.edu

## Abstract

Longitudinal biomedical studies monitor individuals over time to capture dynamics in brain development, disease progression, and treatment effects. However, estimating trajectories of brain biomarkers is challenging due to biological variability, inconsistencies in measurement protocols (e.g., differences in MRI scanners) as well as scarcity and irregularity in longitudinal measurements. Herein, we introduce a novel personalized deep kernel regression framework for forecasting brain biomarkers, with application to regional volumetric measurements. Our approach integrates two key components: a *population* model that captures brain trajectories from a large and diverse cohort, and a *subject-specific* model that captures individual trajectories. To optimally combine these, we propose *Adaptive Shrinkage Estimation*, which effectively balances population and subject-specific models. We assess our model's performance through predictive accuracy metrics, uncertainty quantification, and validation against external clinical studies. Benchmarking against state-of-the-art statistical and machine learning models—including linear mixed effects models, generalized additive models, and deep learning methods—demonstrates the superior predictive performance of our approach. Additionally, we apply our method to predict trajectories of composite neuroimaging biomarkers, which highlights the versatility of our approach in modeling the progression of longitudinal neuroimaging biomarkers. Furthermore, validation on three external neuroimaging studies confirms the robustness of our method across different clinical contexts. We make the code available at https://github.com/vatass/AdaptiveShrinkageDKGP.

## 1 Introduction

Accurately predicting the temporal progression of brain biomarkers is essential for monitoring disease progression and determining optimal intervention points (Maheux et al., 2023). However, challenges such as biological variability among individuals, limited longitudinal data, and irregular observation intervals make model development particularly difficult. Since accurate and reliable predictions are imperative, models must dynamically adapt as new subject-specific data become available, ensuring personalized predictions.

Several predictive models have been proposed to model the progression of biomarkers in the field of neuroimaging (Marinescu et al., 2018). Traditional methods, such as linear mixed effects models (Lindstrom & Bates, 1988), often struggle to handle high-dimensional multivariate data effectively and are predominantly used for statistical inference (Bernal-Rusiel et al., 2013; Xie et al., 2023). Additionally, mixed-effect regression modeling is commonly employed to address longitudinal predictions by fitting biomarker progression to linear or sigmoidal curves (Sabuncu et al., 2014; Koval et al., 2021). However, this approach may be limited by its reliance on predefined trajectory shapes. More recently, Hong et al. (2019) and Gruffaz et al. (2021) explored manifold learning techniques to capture biomarker trajectories requiring subjects with at least two acquisitions for inference. Additionally, Lorenzi et al. (2019) introduced a Gaussian process–based disease progression model capable of predicting biomarkers like cognitive scores and volumetric measurements, but it relies

on specific design assumptions regarding the number of observations per subject and also uses low-dimensional input (i.e., five biomarkers). In the same spectrum, Koval et al. (2021) presented a Bayesian mixed effects model for estimating biomarker trajectories from low-dimensional inputs. Abi Nader et al. (2020) proposed a method for spatiotemporal progression of biomarkers without adapting to subject's follow-up. Tassopoulou et al. (2022) proposed a deep kernel regression method to infer biomarker trajectories from high-dimensional multivariate imaging features, though it does not utilize individual subject trajectories to refine predictions. In a related direction, Rudovic et al. (2019) developed a meta-weighting scheme combining two personalized Gaussian process models to forecast ADAS-Cog13 (Mohs et al., 1997) scores up to two years ahead. Similarly, Chung et al. (2019) introduced a deep mixed effects framework for personalization in electronic health record time-series data, employing a long short-term memory network (Hochreiter & Schmidhuber, 1997) to model population trends while using a Gaussian process to capture subject-specific deviations.

In this paper, we address the above limitations by proposing *Deep Kernel Regression with Adaptive Shrinkage Estimation*, a composite framework for predicting longitudinal brain trajectories leveraging all the available observations of the test subject, either single acquisition or multiple randomly-timed acquisitions. Unlike previous approaches that predict biomarkers within predetermined time intervals (Rudovic et al., 2019), our method is designed to forecast over a practically unbounded future time horizon while simultaneously refining past observations by reducing noise in subject-specific observations. This dual capability enhances measurement reliability and preserves the global progression trend from the initial observation to any unseen future time point. Moreover, our framework naturally handles randomly-timed and temporally unaligned longitudinal observations without requiring imputation, thereby leveraging all available data. By extending the shrinkage estimator concept from Bayesian statistics and penalized inference (James & Stein, 1961; Shou et al., 2014), our method learns weights to combine population and subject-specific deep kernel model through an adaptive shrinkage estimator, while accounting for both observation time and predictive uncertainty.

**Contributions. 1)** We propose a novel deep kernel regression framework for predicting biomarker trajectories from sparse longitudinal observations, that maps high-dimensional, imaging and clinical features into a lower-dimensional latent space predictive of biomarker progression. Our approach naturally accommodates randomly-timed and temporally unaligned observations without requiring imputation. **2)** We introduce Adaptive Shrinkage Estimation that fuses the population and subject-specific models. This framework enables incremental updates to personalized predictions as new data arrive and it also refines historical observations to reduce noise while preserving the overall progression trend from the first observation to any future time. Importantly, the Adaptive Shrinkage estimator is interpretable, offering insights into the relative contributions of population and subject-specific model. **3)** We showcase the versatility of our method to be applied for the prediction of two additional composite neuroimaging biomarkers from high-dimensional multivariate imaging data and clinical covariates. **4)** We demonstrate the generalizability of our method in different clinical contexts, showing its ability to generalize in three external clinical studies.

## 2 METHOD

### 2.1 PROBLEM FORMULATION

We address the problem of predicting biomarker trajectories, modeled as a one-dimensional signal spanning multiple years. Formally, biomarker progression is described by the function $f : U \to Y$, where $U \in \mathbb{R}^K$ and $Y \in \mathbb{R}$. The input is represented as $\mathbf{U} = (X, M, T)$, where $X$ denotes the imaging features, $M$ denotes the clinical covariates at subject's first visit, and $T$ represents the temporal variable, indicating time in months from the first visit. The biomarker trajectory is denoted as $Y = (y_0, y_1, \ldots, y_n)$, corresponding to the biomarker values at time points $T = (t_0, t_1, \ldots, t_n)$. Our goal is to learn smooth functions biomarker progression using imaging and clinical data. To achieve this, we employ Deep Kernel Learning (DKL) (Wilson et al., 2015). The deep kernel integrates imaging and clinical covariates, learning a lower-dimensional representation informative for biomarker progression, while a Gaussian Process (GP) models the temporal dependencies. The backbone model, Deep Kernel Gaussian Process (DKGP), is defined as: $f(\mathbf{U}) \sim \mathcal{GP}(\mu, \mathbf{K}(\Phi(\mathbf{U}), \Phi(\mathbf{U})))$, where $\Phi$ is a transformation function.

### 2.2 POPULATION DEEP KERNEL MODEL (P-DKGP)

The population model leverages data from the population dataset $D_p = \{\mathbf{U}_p, \mathbf{Y}_p\}$, comprising subjects with longitudinal observations. It applies the transformation $\Phi(u; \mathbf{W}, \mathbf{b})$, a Multi-Layer

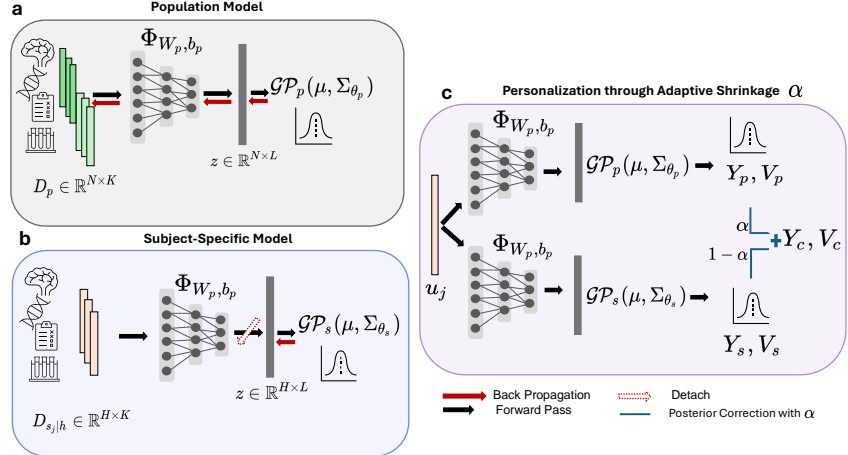

Figure 1: Overview of the proposed framework. In Figure 1**a**, we illustrate the training process of the two models, p-DKGP. The population dataset $D_p$ contains multiple longitudinal acquisitions of subjects, where $N$ is the total number of samples across all subjects, and $L$ is the latent dimension obtained from transformation $\Phi$. Different shades of green in the population dataset indicate different subjects in $D_p$. In Figure 1**b**, we illustrate the training process of the ss-DKGP. We denote the observed trajectory of subject $j$ with $h$ samples as $D_{s_j|h}$. These samples are utilized to train the ss-DKGP. During the training of the ss-DKGP, the transformation $\Phi$ is fixed, and only the subject-specific Gaussian process is optimized. In Figure 1**C**, we visualize the personalization process through the adaptive shrinkage parameter $\alpha$. For subject $j$, we extrapolate biomarker values over time using both the p-DKGP and ss-DKGP models. These extrapolated values are then used to infer the adaptive shrinkage $\alpha$ for posterior correction, yielding the personalized posterior predictive mean $Y_c$ variance $V_c$ of the subject's trajectory.

Perceptron (MLP), that maps the input data $\mathbf{U}_p = (X, M, T)$ into a latent representation:

$$\mathbf{Z}_p = \Phi(\mathbf{U}_p; \mathbf{W}, \mathbf{b}). \tag{1}$$

A GP, subsequently, models the biomarker progression function $f$ using a Radial Basis Function (RBF) kernel as the covariance function and a zero mean: $f(\mathbf{Z}_p) \sim \mathcal{GP}\left(0, K(\mathbf{Z}_p, \mathbf{Z}_p')\right)$.

The population parameters $\gamma_p = \{\mathbf{W}_p, \mathbf{b}_p, l_p, \sigma_p^2, \sigma_{n_p}^2\}$ include both the transformation parameters of $\Phi$ and the Gaussian Process (GP) hyperparameters: the lengthscale $l_p$, signal variance $\sigma_p^2$, and noise variance $\sigma_{n_p}^2$. These parameters are jointly learned by maximizing the Marginal Log Likelihood (MLL) of the GP (Wilson et al., 2015; Rasmussen & Williams, 2006).

For a test subject $j$ with input $u_j = (x_j, m_j, t)$, we denote the transformed input as $z_j = \Phi(u_j; \mathbf{W}_p, \mathbf{b}_p)$.

The posterior predictive distribution of the biomarker function at point $u_j = (x_j, m_j, t)$ is:

$$f_{p_j} \mid (\mathbf{Z}_p, \mathbf{Y}_p), z_j \sim \mathcal{N}(\bar{\mathbf{f}}_{p_j}, \mathrm{cov}(\mathbf{f}_{p_j})). \tag{2}$$

The mean and variance of the predictive posterior distribution provide the predictions and their uncertainties, respectively, and are calculated as follows:

$$\bar{\mathbf{f}}_{p_j} = \mathbb{E}[\mathbf{f}_* \mid \mathbf{Z}_p, \mathbf{Y}_p, z_j] = K(z_j, \mathbf{Z}_p)[K(\mathbf{Z}_p, \mathbf{Z}_p) + \sigma_{n_p}^2 I]^{-1}\mathbf{Y}_p, \tag{3}$$

$$\mathrm{Var}(\mathbf{f}_{p_j}) = K(z_j, z_j) - K(z_j, \mathbf{Z}_p)[K(\mathbf{Z}_p, \mathbf{Z}_p) + \sigma_{n_p}^2 I]^{-1}K(\mathbf{Z}_p, z_j), \tag{4}$$

where $\sigma_{n_p}^2$ is the additive independent identically distributed Gaussian noise $\epsilon$.

For simplicity, the predictive mean and variance of a biomarker for test subject $j$ from the p-DKGP are denoted as $y_p$ and $v_p$, respectively. By prompting the p-DKGP model with different time intervals $t$, yields the predicted trajectory and predictive uncertainty across time, represented as $Y_p = (y_{p_1}, y_{p_2}, \ldots, y_{p_T})$ and $V_p = (v_{p_1}, v_{p_2}, \ldots, v_{p_T})$.

## 2.3 SUBJECT-SPECIFIC DEEP KERNEL MODEL (SS-DKGP)

For a new test subject, let $h$ denote the number of observations and $T_{obs}$ the time of observation from the initial acquisition. The observed data for the subject is represented as $D_s = \{(X_s, M_s, T_s), Y_s\}$. The ss-DKGP model is trained on $D_s$ to capture the subject-specific trajectory. The transformation $\Phi(\cdot; \mathbf{W}_p, \mathbf{b}_p)$, learned via the p-DKGP, initializes the deep kernel of the subject-specific model.

We initialize a new GP with an RBF kernel and a zero mean. During the training of the ss-DKGP, only the observed trajectory of the subject is used. Specifically, we update the GP hyperparameters, which include the lengthscale $l_s$ and the signal variance $\sigma_s$, while keeping the weights of the function $\Phi(\cdot; \mathbf{W}_p, \mathbf{b}_p)$ frozen during backpropagation. The subject-specific GP hyperparameters $\gamma_s = \{l_s, \sigma_s^2, \sigma_{n_s}^2\}$ are jointly learned by maximizing the MLL of the GP.

For subject $j$ with input $u_j = (x_j, m_j, t)$, we denote their transformation as $z_j = \Phi(u_j; \mathbf{W}_p, \mathbf{b}_p)$.

The posterior predictive distribution of the biomarker progression function at time point $t$ is:

$$f_{s_j} \mid (\mathbf{Z}_s, \mathbf{Y}_s), z_j \sim \mathcal{N}(\bar{\mathbf{f}}_{s_j}, \text{cov}(\mathbf{f}_{s_j})), \tag{5}$$

where $z_j = \Phi(u_j; \mathbf{W}_p, \mathbf{b}_p)$

The predictive mean and variance, representing the predictions and their associated uncertainties respectively, are computed as follows:

$$\bar{\mathbf{f}}_{s_j} = \mathbb{E}[\mathbf{f}_{s_j} \mid \mathbf{Z}_s, \mathbf{Y}_s, z_j] = K(z_j, \mathbf{Z}_s)[K(\mathbf{Z}_s, \mathbf{Z}_s) + \sigma_{n_s}^2 I]^{-1}\mathbf{Y}_s, \tag{6}$$

$$\text{Var}(\mathbf{f}_{s_j}) = K(z_j, z_j) - K(z_j, \mathbf{Z}_s)[K(\mathbf{Z}_s, \mathbf{Z}_s) + \sigma_{n_s}^2 I]^{-1}K(\mathbf{Z}_s, z_j). \tag{7}$$

where $\sigma_{n_s}^2$ is the additive independent identically distributed Gaussian noise $\epsilon$.

For simplicity, the predictive mean and predictive variance of the ss-DKGP are denoted as $y_{sj}$ and $v_{sj}$, respectively. By querying the ss-DKGP model at different time intervals $t$ we reconstruct the biomarker trajectory of subject $j$, yielding the predicted trajectory $Y_s = (y_{s_1}, y_{s_2}, \ldots, y_{s_T})$ and predictive uncertainty $V_s = (v_{s_1}, v_{s_2}, \ldots, v_{s_T})$.

## 2.4 PREDICTIVE POSTERIOR CORRECTION

Given predictions $y_p$ and $y_s$ from the p-DKGP and ss-DKGP models, the personalized prediction is expressed as a linear combination:

$$y_c = \alpha y_p + (1 - \alpha)y_s, \tag{8}$$

where, $\alpha$ is the shrinkage parameter reflecting the relative confidence in each model. Assuming independence between the models, the combined prediction $y_c$ retains Gaussian properties, and its variance is given by:

$$v_c = \alpha^2 v_p + (1 - \alpha)^2 v_s. \tag{9}$$

In Supplementary Section 2.4 we address the independence assumption and its impact.

The weights $\alpha$ and $1 - \alpha$ quantify the credibility of each model, yielding a new posterior predictive mean $Y_c$ and variance $V_c$. Values of $\alpha$ close to 1 indicate higher confidence in p-DKGP model, while values close to 0 reflect greater trust in ss-DKGP model. We refer to $\alpha$ as the shrinkage parameter.

### 2.4.1 ACQUIRING THE ORACLE SHRINKAGE $\alpha$

Estimating the oracle shrinkage parameter $\alpha$ is crucial for constructing the personalized posterior predictive means and variances of the biomarker trajectory. To estimate $\alpha$, we use a held-out set of subjects with known trajectories, unseen by the population model. Predictions for these subjects are generated using the p-DKGP model. For each subject, the ss-DKGP component is trained by progressively increasing the length of the observed trajectory.

The entire biomarker trajectory is reconstructed from the baseline time ($t = 0$) to the subject's last time point $t_n$. Using both models, p-DKGP and ss-DKGP, we obtain two estimates of the biomarker trajectory along with their predictive variances. Let $Y_p$ and $V_p$ denote the p-DKGP predictive mean and variance, and $Y_s$ and $V_s$ denote the ss-DKGP model predictive mean and variance. Let $Y$ represent the ground truth biomarker values over time. The oracle $\alpha$ is estimated by minimizing the following criterion:

$$J_{s|h}(\alpha) = \sum_{t=0}^{t_n} (y_t - (\alpha \cdot y_{p_t} + (1 - \alpha) \cdot y_{s_t}))^2. \tag{10}$$

The notation $J_{s|h}$ reflects that this optimization is performed for a subject $s$, given $h$ observed acquisitions. The algorithm for calculating the oracle shrinkage estimates on the validation set is outlined in Algorithm 1. Each subject's data is processed individually, applying the optimization to each sequence of observations. This process is repeated for every subject in the validation set.

---

**Algorithm 1** Oracle Shrinkage Estimation

---

**Require:** Validation set $V = \{(U^s, Y^{(s)}) \mid s \in S\}$, where $Y^{(s)} = \{y_t^{(s)}\}_{t=1}^T$ is the ground truth trajectory for subject $s$
**Ensure:** Optimal shrinkage parameters $\hat{\alpha}_{s,h}$ for each $s \in S$ and $h \in H$
 1: **for** each $s \in S$ **do**
 2:      Initialize list $L^{(s)} \leftarrow [\,]$
 3:      **for** each $h \in H$ **do**
 4:          Obtain P-DKGP trajectory: $Y_p^{(s,h)} = \{y_{p,t}^{(s,h)}\}_{t=1}^T$
 5:          Obtain ss-DKGP trajectory: $Y_s^{(s,h)} = \{y_{s,t}^{(s,h)}\}_{t=1}^T$
 6:          Define objective function:

$$J_{s,h}(\alpha) = \sum_{t=0}^{T} \left(y_t^{(s)} - \left(\alpha y_{p,t}^{(s,h)} + (1-\alpha)y_{s,t}^{(s,h)}\right)\right)^2$$

 7:          Compute:

$$\hat{\alpha}_{s,h} = \arg\min_{\alpha \in [0,1]} J_{s,h}(\alpha)$$

 8:          Append $\hat{\alpha}_{s,h}$ to $L^{(s)}$
 9:      **end for**
10:      Store list $L^{(s)}$ for subject $s$
11: **end for**

---

### 2.4.2 LEARNING THE ADAPTIVE SHRINKAGE $\alpha$

The shrinkage parameter $\alpha$ represents the trust factor between the two components (p-DKGP and ss-DKGP). We model $\alpha$ as a function of the input variables $q = \{y_p, y_s, v_p, v_s, T_{\text{obs}}\}$, where $q \in \mathbb{R}^5$ and $T_{\text{obs}}$ represents the time of observation. Using oracle shrinkage $\alpha$ obtained from Section 2.4.1 on the validation set, our objective is to learn a mapping function $g_a$ that transforms the input space $q \in \mathbb{R}^5$, to the output space of adaptive shrinkage $\alpha \in \mathbb{R}$, as $\hat{\alpha} = g_a(q; \theta)$.

We employ XGBoost regression to learn the function $g$ that minimizes the difference between the predicted $\hat{\alpha}$ and the oracle $\alpha$. The learned function is denoted as $g_\alpha$. In Supplementary Section C.2, we provide results from additional non-linear functions we experiment with, demonstrating that XGBoost achieves the best performance for estimating the shrinkage $\alpha$.

### 2.5 PERSONALIZATION THROUGH ADAPTIVE SHRINKAGE ESTIMATION

For a new test subject with $h$ observations and $T_{\text{obs}}$ as the observation time (measured from the subject's first visit), we train the ss-DKGP model as described in Section 2.3. The posterior-corrected predictive distribution, referred to as pers-DKGP, is computed using the following algorithm:

---

**Algorithm 2** Personalization through Adaptive Shrinkage Estimation

---

**Require:** p-DKGP model, ss-DKGP model, and learned function $g_\alpha$
**Ensure:** Adapted predictive mean and variance: $Y_c, V_c$
 1: Compute $Y_p, V_p$ (predictive mean and variance) from the p-DKGP model.
 2: Compute $Y_s, V_s$ (predictive mean and variance) from the ss-DKGP model.
 3: Adapted Shrinkage Estimation: $\hat{\alpha}_h = g_\alpha(Y_p, Y_s, V_p, V_s, T_{\text{obs}})$.
 4: Compute the personalized predictive mean: $Y_c = \hat{\alpha}_h \cdot Y_p + (1 - \hat{\alpha}_h) \cdot Y_s$.
 5: Compute the personalized predictive variance: $V_c = \hat{\alpha}_h^2 \cdot V_p + (1 - \hat{\alpha}_h)^2 \cdot V_s$.
 6: **return** $Y_c, V_c$.

---

The personalization process through Adaptive Shrinkage Estimation is described in Algorithm 2.

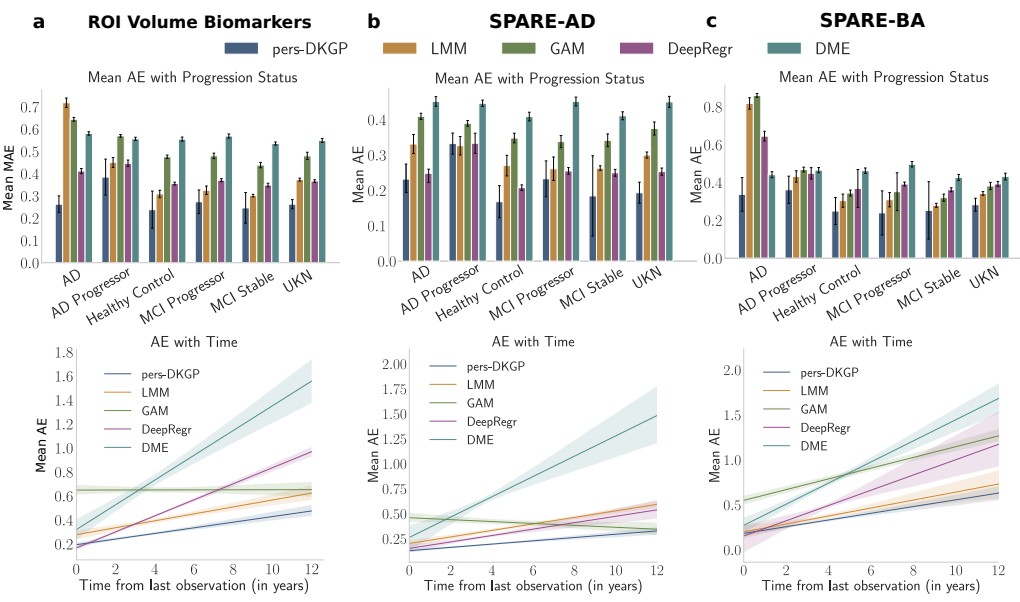

Figure 2: We compare the mean MAE per subject stratified by the progression status (top) and the AE with time from the last observation (bottom) of our method with the baselines for (a) the 7 ROI Volume biomarkers, (b) SPARE-AD score and (c) SPARE-BA. Error bars, in the top row, denote the 95th percentile of the MAE across all subjects. Our method is denoted as pers-DKGP.

## 3 EXPERIMENTS

### 3.1 PREDICTION OF REGIONAL VOLUMETRIC TRAJECTORIES

In this section, we apply deep kernel regression with Adaptive Shrinkage Estimation to predict trajectories of seven volumetric Regions of Interest (ROI): Hippocampus R, Hippocampus L, Thalamus Proper R, Amygdala R, Amygdala L, Parahippocampal Gyrus R and Lateral Ventricle R. For each ROI Volume model we use a dataset of $2,200$ subjects with $U_i = (X_i, M_i, T_i)$ from subject $i$, where $X_i$ are volumetric measures from 145 brain regions collected at subject's first visit, $M_i$ are the covariates of diagnosis at subject's first visit, sex, age, education, APOE4 Alleles, a genetic variant related to AD and $T_i$ is the time from subject's first visit. For each ROI Volume biomarker, the p-DKGP model is trained on a population cohort of $1,600$ subjects, while the adaptive shrinkage estimator is trained on a held-out set of 200 subjects. Predictive performance is evaluated on $440$ test subjects. For details on the architecture and training of the ROI Volume deep kernel models, p-DKGP and ss-DKGP, see Section B.1.

We combine preprocessed and harmonized neuroimaging measures from two well-known longitudinal studies: the Alzheimer's Disease Neuroimaging Initiative (ADNI) (Weiner et al., 2017) and the Baltimore Longitudinal Study of Aging (BLSA) (Ferrucci, 2008), which focus on Alzheimer's Disease and Brain Aging, respectively. Further details on the studies and preprocessing pipelines are provided in Supplementary Section A.

We benchmark our method against several baselines and state-of-the-art predictors: Linear Mixed Model (LMM) (Lindstrom & Bates, 1988), Generalized Additive Model (GAM) (Hastie & Tibshirani, 1986), Deep Neural Network Regression, and the Deep Mixed Effects (DME) (Chung et al., 2019). Further details on the architectural design and training of baselines are provided in Supplementary B.3. Model performance is evaluated from two perspectives: predictive accuracy and uncertainty quantification (UQ). Predictive accuracy is measured using Absolute Error (AE) and Mean Absolute Error (MAE) per subject. UQ is assessed by interval width (the range between ±2 standard deviations from the predictive mean) and coverage (the proportion of true biomarker values within that range). Importantly, these metrics are computed over the entire unseen trajectory of test subjects, providing a comprehensive evaluation of model performance over time. We refer to our method as personalized-DKGP or shortly pers-DKGP.

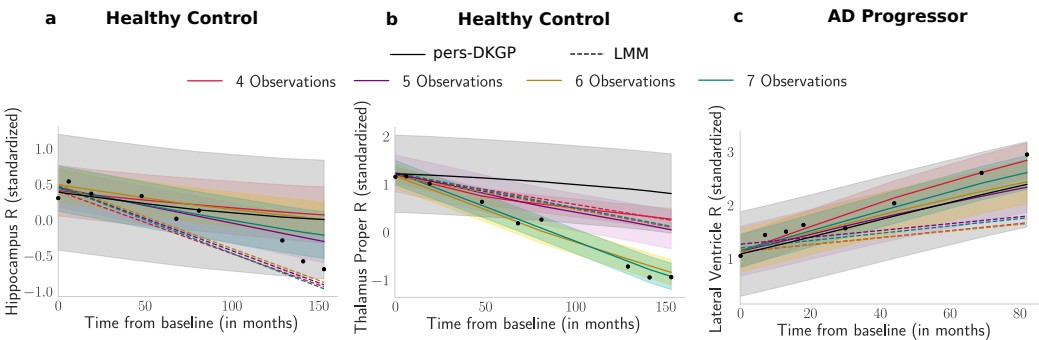

Figure 3: We present personalized ROI volume trajectories for three test subjects as observations increase from 4 to 7 acquisitions. The dashed lines represent the prediction using LMM. The first two panels visualize the Hippocampus R and Thalamus Proper R Volume trajectories of Healthy Control subject. Last panel shows the Lateral Ventricle R Volume for an AD Progressor. The shaded bands represent the predictive uncertainty over time.

For each predictor, Figure 2 **a** presents a comparative study of the predictive performance with respect to progression status and time from the last known acquisition. Progression status is defined by the subject's initial and final diagnoses, categorized as follows: AD refers to subjects diagnosed with AD at their first visit; AD Progressor includes subjects initially diagnosed as Cognitively Normal (CN) or Mild Cognitively Impaired (MCI) who progress to AD; Healthy Controls are subjects who remain CN throughout all visits; MCI Progressor refers to subjects who progress from CN to MCI; MCI Stable includes subjects who remain MCI throughout their trajectory; and Unknown (UKN) corresponds to cases involving misdiagnosis.

Building on this categorization, Figure 2**a** shows the mean MAE across progression status for the seven volumetric ROIs. Notably, the largest mean MAE differences between our method and baselines occur in participants with AD and AD Progressors, who exhibit non-linear and steeper trends that competing baselines fail to capture. Specifically, the Linear Mixed Model (LMM), constrained to linear patterns in ROI volumes, shows significant percentage mean MAE differences in AD (177.66%) and AD progressors (22.05%). Even in healthy controls, LMM exhibits a 29.78% MAE difference, highlighting its inability to capture trajectories even in cases of relatively stable volume trajectories. Further quantitative comparisons, including error stratification by covariates such as sex, APOE4 Alleles, and education years, are provided in Supplementary Sections D.1.

In addition to evaluating performance with respect to progression status, we also assess the model's ability to predict long-term longitudinal trajectories. In Figure 2**a**, we visualize the mean AE across different lengths of observed trajectories, with errors plotted relative to the time from the last observation. Our method achieves progressively lower mean AE over time, indicating improved precision in both long-term and short-term predictions. This demonstrates the model's ability in capturing temporal trends and adapting to varying observation lengths.

To further highlight the strengths of our model, we provide a qualitative evaluation of the predicted trajectories in Figure 3. For the Volume ROIs of Hippocampus R, Thalamus Proper R, and Lateral Ventricle R, our model successfully adapts to the observations of test subjects, resulting in more accurate long-term predictions. For instance, in the Healthy Control subject shown in Figure 3**b**, the population prediction deviates from the actual trajectory. However, as the number of observations increases, the pers-DKGP trajectory shifts toward the observed trajectory, effectively adapting to the subject-specific trend. Similarly, the third subject, an AD Progressor in Figure 3**c**, exhibits an abrupt increase in ventricular volume. This trend is captured with few observations by the pers-DKGP model, while the LMM underestimates the ventricular volume in the long term. Additional qualitative examples of trajectories are provided in Supplementary Section D.3.

Overall, the LMM exhibits limited flexibility in capturing non-linear patterns in ROI volumes, rendering it inadequate for long-term biomarker prediction. While it performs reasonably well in short-term forecasts and lower-dimensional settings, its expressiveness falls short for complex, high-dimensional inputs. Deep regression, though capable of learning from observed data, often

yields non-smooth or non-monotonic trajectories that deviate from biologically plausible biomarker progression trends. The DME model, which combines a shared deep mean function with subject-specific GP, struggles to achieve personalization in high-dimensional input spaces, resulting in persistent errors across time and diagnostic categories. These issues stem from the limitations of the RBF kernel in managing multivariate, high-dimensional data. In contrast, our method effectively approximates non-linear mixed effects models, demonstrating flexibility in handling multivariate, high-dimensional data and capturing diverse temporal patterns.

## 3.2 APPLICATION TO NEUROIMAGING BIOMARKERS: SPARE SCORES

Having demonstrated our framework's ability to personalize longitudinal predictions of volumetric ROIs as subject observations increase, we now show its versatility by applying it to composite neuroimaging biomarkers: the SPARE-AD (Davatzikos et al., 2009) and SPARE-BA (Habes et al., 2016) scores. SPARE-AD quantifies the risk of AD progression, while SPARE-BA represents predicted brain age. For both SPARE models we use a dataset of 2,200 subjects with $U_i = (X_i, M_i, T_i)$ from subject $i$, where $X_i$ are volumetric measures from 145 brain regions collected at subject's first visit, $M_i$ are the covariates of diagnosis at subject's first visit, sex, age, education, APOE4 Alleles, the SPARE-AD and SPARE-BA values at the first visit and $T_i$ is the time from subject's first visit. The p-DKGP model is trained on 1600 subjects, the adaptive shrinkage estimator is trained on a held-out set of 200 subjects. The evaluation of the predictive performance is performed on the 440 test subjects. For details on the architectural design and training of the SPARE-AD and SPARE-BA deep kernel models (p-DKGP and ss-DKGP) see Section B.2.

Our model demonstrates strong performance in predicting long-term longitudinal trajectories for both SPARE-AD and SPARE-BA biomarkers, as illustrated in Figure 2**b** and 2**c**. Notably, the model achieves progressively lower mean AE over time, indicating improved precision in forecasting long-term outcomes. For SPARE-BA, model performance differences are minimal in stable subjects and healthy controls, but more pronounced in AD subjects, where SPARE-BA exhibits steeper progression trends due to accelerated brain aging. For the SPARE-AD biomarker, we also visualize absolute error with the number of observations. This highlights how our model adapts with increasing observations, starting with a single scan using the p-DKGP model ($\alpha = 1$) and transitioning to adapted shrinkage estimation for personalization as follow-up observations increase. Evidence is provided in Table 5 and Figure 7 in Supplementary Section D.2.

## 3.3 APPLICATION TO EXTERNAL NEUROIMAGING STUDIES

In this section, we demonstrate the generalizability of our method to previously unseen neuroimaging datasets. After training the p-DKGP and adaptive shrinkage estimator on the population and validation datasets from the ADNI and BLSA cohorts, we personalize starting from the first follow-up point for each subject and predict the remaining trajectory. This process is repeated for all follow-up points, with the very last follow-up reserved for testing.

We test the performance of our framework on subjects from three independent clinical studies: OASIS (Marcus et al., 2010), AIBL (Ellis et al., 2009), and PreventAD (Tremblay-Mercier et al., 2021). These datasets differ from the training population in terms of demographics, diagnosis composition, and follow-up intervals, presenting a challenging test of the model's generalizability across diverse populations. In Supplementary Section A we present details on the demographic and clinical characteristics of these studies.

The three external studies exhibit notable differences in demographics and follow-up intervals: **AIBL**: Includes 82 individuals with a mean age of 75 years, which is close to the mean age of the joint cohort of ADNI and BLSA. It is predominantly composed of AD patients followed by MCI and Healthy Controls. On average, each subject has approximately 3 follow-up visits, with a mean interval of 24 months between visits. **OASIS**: Includes 559 individuals younger on average (67.8 years) compared to both ADNI and BLSA. It is primarily composed of healthy controls, with smaller representations of MCI and AD cases. The average number of follow-ups is $\sim 3$ per subject, with a mean interval of 32 months. **PreventAD**: Includes 271 individuals and focuses on pre-symptomatic early detection of AD in a healthier and younger population (mean age 65.3 years) with an average of 4 follow-up visits per subject and a shorter mean interval of 10 months.

Our method outperforms baseline predictors across three independent clinical studies—AIBL, OASIS, and PreventAD—underscoring its effectiveness in diverse, real-world scenarios (Figure 4). The model achieves lower MAE compared to baselines, with narrow confidence intervals reflecting its

stability. In the AIBL study, pers-DKGP achieves a Mean AE of *0.197 ± 0.009*, substantially outperforming the baseline methods. A similar trend is observed in the OASIS study, where pers-DKGP attains a Mean AE of *0.259 ± 0.006*. Notably, in the PreventAD study, our method achieves the lowest Mean AE of *0.139 ± 0.004*, outperforming LMM and GAM. The narrow CIs of the AE associated with pers-DKGP across all datasets highlight its reliability and consistent precision, even in the presence of data variability. Interestingly, the lowest error observed in the PreventAD study, along with the reduced disparity between pers-DKGP and statistical models like LMM and GAM, is attributed to the younger population and shorter follow-up intervals in this dataset. Predicting Volume ROIs in a younger, healthier control population, as in PreventAD, is inherently less challenging compared to the older, partially demented populations in OASIS and AIBL.

Collectively, these results position our model as a robust and reliable framework for personalized forecasting of neuroimaging biomarkers, offering potential for application in clinical trials and neuroimaging studies.

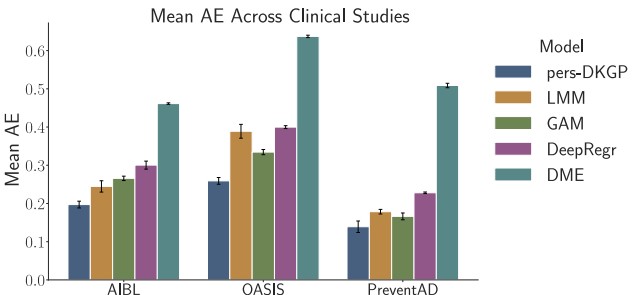

Figure 4: We evaluate the mean absolute error for the seven ROI Volume biomarkers across three external neuroimaging studies. Error bars denote the 95th percentile of the absolute error. Notice that the pers-DKGP achieves the lowest error across all external studies, in comparison with the competing baselines.

### 3.4 EXPLAINING ADAPTIVE SHRINKAGE: AN ABLATION STUDY ON THE $\alpha$ ESTIMATOR

In this section, we demonstrate the effectiveness and interpretability of the Adaptive Shrinkage estimator. We first compare it to alternative posterior correction approaches and then use explainability analysis to illustrate how Adaptive Shrinkage estimator learns to balance the two posterior predictive distributions in a data-driven manner, making its decision-making process intuitive.

We explore various strategies for selecting the shrinkage parameter $\alpha$. First, we experiment with a constant $\alpha = c$, where $c \in (0, 1)$, representing an uninformative approach to posterior correction. Next, we employ a semi-informative (deterministic) approach, where the $\alpha$ for each test subject is determined by optimizing the objective in Equation 10 using only subject's observed trajectory. Finally, we use Adaptive Shrinkage estimator to determine $\alpha$. We conduct this experiment for seven ROI Volume biomarkers: Hippocampus R/L, Lateral Ventricle, Thalamus Proper, Amygdala R/L, and the Parahippocampal Gyrus R. Here, we present results for Hippocampus R, Lateral Ventricle, and Thalamus Proper under the constant $\alpha$ and Adaptive Shrinkage. Results for the remaining Volume ROIs and the deterministic approach are provided in Table 6 of Supplementary Section D.4.1.

The deterministic approach (Table 6) results in the worst outcomes in terms of both predictive performance and uncertainty quantification, suggesting that the observed trajectory alone is insufficient to determine the $\alpha$ for future predictions. Per-subject optimized $\alpha$ can overfit the noise in a single subject's limited data, leading to poorer generalization, whereas the learned adaptive shrinkage generalizes better across subjects. Additionally, in the constant $\alpha$ section of Table 1, we present the performance of the best constant $\alpha$ values. This demonstrates that optimal performance is not achieved through simple averaging and that the optimal $\alpha$ varies significantly across ROIs. For example, the best $\alpha$ is 0.5 for Hippocampus, 0.3 for Lateral Ventricle, and 0.7 for Thalamus Proper. These results highlight the inadequacy of a one-size-fits-all approach and underscore the necessity for a more sophisticated method. The evidence suggests that Adaptive Shrinkage provides a more informed approach for determining the ideal $\alpha$, leading to improved predictive performance and uncertainty quantification.

Table 1: Ablation study on the shrinkage parameter $\alpha$. We report the Mean AE along with its 95% percentile CI, Mean Coverage, and Mean Interval Width

| ROI | $\alpha$ | Best Constant | | | Adaptive Shrinkage | | |
|---|---|---|---|---|---|---|---|
| | | Mean AE (CI) | Mean Cov. | Mean Int. | Mean AE (CI) | Mean Cov. | Mean Int. |
| Hippocampus R | 0.5 | 0.257 (±0.007) | 0.808 | 0.843 | **0.243** (±0.003) | 0.795 | 0.902 |
| Lateral Ventricle R | 0.3 | 0.143 (±0.006) | 0.853 | 0.507 | **0.131** (±0.002) | 0.855 | 0.626 |
| Thalamus Proper R | 0.7 | 0.241 (±0.007) | 0.934 | 1.127 | **0.219** (±0.003) | 0.849 | 0.911 |

Following, we elucidate the decision-making process of Adaptive Shrinkage with explainability analysis. We focus on the impact of each input variable—$Y_p$, $Y_s$, $V_p$, $V_s$, and $T_{obs}$—and their interactions on the prediction of the adaptive shrinkage parameter $\alpha$. Specifically, we aim to understand how the deviation between the population and subject-specific predictive means ($\delta_y = Y_p - Y_s$) and the observation time $T_{obs}$ influence the model's predictions.

We employ SHAP (SHapley Additive exPlanations) values (Lundberg & Lee, 2017) to interpret the contribution of each feature to individual predictions. Figure 13 in Supplementary Section D.4 reveals that $T_{obs}$ is the most influential variable in the decision-making process. This is further validated by the observation that the distribution of adaptive shrinkage $\alpha$ decreases as the number of follow-up visits (and thus $T_{obs}$) increases. Figure 12 in Supplementary Section D.4 demonstrates the distribution of $\alpha$ with the number of observations for the seven ROIs and SPARE scores, as well as the adaptive shrinkage $\alpha$ obtained from external neuroimaging studies. The consistent trend of decreasing $\alpha$ as the number of observations increases highlights the biomarker-agnostic ability of Adaptive Shrinkage to optimally combine population and subject-specific trends. This behavior is also consistent across external neuroimaging studies, further validating the generalizability of the approach. Additional qualitative results demonstrating the decision-making process of Adaptive Shrinkage are provided in Supplementary Section D.3, Figures 10 and 11.

Furthermore, correlation analysis (Supplementary Section D.4, Table 7) reveals a consistent negative relationship between $T_{obs}$ and the predicted $\alpha$ when the deviation $\delta_y$ is large. This indicates that, in the presence of significant deviations between the two predictors, Adaptive Shrinkage reduces the weight assigned to the population-level model (p-DKGP) for longer observation periods. This aligns with the intuition that as more follow-up observations are available, greater trust is placed in the subject-specific predictive distribution.

## 4 DISCUSSION

In this paper, we introduce deep kernel regression with Adaptive Shrinkage Estimation for predicting personalized biomarker trajectories via posterior correction. Our method learns the adaptive shrinkage parameter that effectively combines two posterior predictive distributions, enabling the predictive trajectory to adapt to each subject's follow-up acquisitions. Additionally, our method is versatile, effectively modeling the progression of longitudinal biomarkers using multivariate imaging data and clinical covariates. Examples of such biomarkers are the cognitive scores (e.g., MMSE, ADAS-Cog13) and blood biomarkers (e.g., Amyloid-$\beta$, Tau protein). Importantly, our approach exhibits generalization capabilities when applied to external neuroimaging studies with diverse demographics and follow-up intervals, which is particularly valuable for real-world applications, where models must perform robustly across heterogeneous populations.

This property is particularly important as the use of predictive models in healthcare is increasingly critical for both patient management and drug development. Cummings et al. (2019) emphasize the need for AI-informed clinical trials, referred to as precision trial design, while Maheux et al. (2023) evaluate predictive models for biomarker trajectories in Alzheimer's Disease, where derived measures—such as the rate of change—serve as quantitative indicators of disease progression during clinical trials. These measures inform decisions on subject inclusion and treatment efficacy, underscoring the importance of reliable and interpretable predictive tools. Our method's adaptive and intuitive design positions it as a valuable tool for clinical trial design, disease progression modeling, treatment effect estimation, and neuroimaging research. By leveraging personalized predicted ROI Volume and neuroimaging biomarkers, such as SPARE-AD, as endpoints for selecting trial subjects, our framework showcases its potential for real-world application.

At the same time, we acknowledge limitations in our approach, particularly the independence assumption between the $\alpha$ parameter and posterior distributions in the posterior correction step (Supplementary Section 2.4). While this simplification impacts uncertainty quantification, it does not affect the posterior-corrected predictive mean, ensuring accurate predictions. Further discussion on this assumption, including its theoretical justification as well as a way to tackle its limitation, is provided in Supplementary Section C. Future work will explore extending Adaptive Shrinkage Estimation to multivariate biomarker trajectories and improving uncertainty quantification in personalized trajectories to address this aforementioned limitation.

**Acknowledgments.** This research is supported by the NIH U24NS130411 RF1AG054409 grant and the NIA contract ZIA-AG000191.

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

APPENDIX

## A  DATASETS AND PREPROCESSING

We use the iSTAGING consortium (Habes et al., 2021) that consolidated and harmonized imaging and clinical data from multiple cohorts. Our real data consists of neuroimaging and demographic measures taken from subjects in the iSTAGING consortium. Specifically, the neuroimaging measures are the 145 anatomical brain ROI volumes (119 ROIs in gray matter, 20 ROIs in white matter and 6 ROIs in ventricles) extracted using a multi-atlas label fusion method (Doshi et al., 2016). Phase-level harmonization was applied on these 145 ROI volumes to remove site effects (Pomponio et al., 2020). Specifically, we use the Alzheimer's Disease Neuroimaging Initiative (ADNI,http://www.adni-info.org/), which is a public-private collaborative longitudinal cohort study and has recruited participants categorized as Cognitively Normal (CN), Mild Cognitive Impairment (MCI) and diagnosed with Alzheimer's Disease (AD) through 4 phases (ADNI1, ADNIGO and ADNI2) (Weiner et al., 2017). We also use Baltimore Longitudinal Study of Aging (BLSA) (Ferrucci, 2008), which has been following participants who are cognitively normal at enrollment with imaging and cognitive exams since 1993.

We also extracted additional studies from the iSTAGING cohort, including the OASIS dataset Marcus et al. (2010), the Australian Imaging, Biomarker, and Lifestyle (AIBL) study (Ellis et al., 2009), and the PreventAD study (Tremblay-Mercier et al., 2021). These studies were exclusively reserved as held-out datasets for evaluating our method on external neuroimaging data.

Our analysis incorporates subjects across all identified progression trajectories: Cognitively Normal (CN) stables, individuals with Mild Cognitive Impairment (MCI), and those progressing to Alzheimer's Disease (AD) from either CN or MCI stages. For the clinical variables, we utilize Age at Baseline, Sex, Years of Education, and APOE4 Allele status, the latter being a known risk factor for Alzheimer's Disease (AD). Diagnostic categories were designated as Cognitively Normal (CN), Mild Cognitive Impairment (MCI), and Alzheimer's Disease (AD). Subjects diagnosed with alternative forms of dementia, such as Lewy Body Dementia and Frontotemporal Dementia, were excluded from the study. These exclusions were minimal and did not significantly impact the overall sample size. Missing diagnostic information was classified as unknown (UKN). Furthermore, Years of Education was dichotomized: subjects with more than 16 years of education were coded as '1', while those with 16 years or fewer were coded as '0'. Detailed demographic and clinical characteristics of the diverse cohort are presented in Table 2.

Table 2: Summary of longitudinal studies with demographic and clinical Information. OASIS, AIBL, and PreventAD studies are used as held-out neuroimaging studies. For age, the mean and the standard deviation are reported. For sex, the number of males and the percentage is presented.

| Study | Subjects | Obs./Subject | #Obs. | Age | Male (%) | Diagnosis (%) | | |
|---|---|---|---|---|---|---|---|---|
| | | | | | | CN | MCI | AD |
| ADNI | 1616 | 5.0±2.0 | 7867 | 73.6±7.0 | 55.5 | 44.7 | 34.6 | 20.6 |
| BLSA | 584 | 3.0±1.0 | 1843 | 74.9±11.1 | 45.7 | 95.8 | 2.8 | 1.4 |
| **OASIS** | 548 | 3.0±1.0 | 1562 | 67.8±9.0 | 42.4 | 88.9 | 1.9 | 12.2 |
| **AIBL** | 82 | 3.0±1.0 | 247 | 75±7.7 | 56.14 | 33.74 | 28.81 | 37.45 |
| **PreventAD** | 271 | 4.2±1.4 | 1141 | 65.3±5.5 | 28.5 | 98.6 | 1.4 | 0.0 |

## B  ARCHITECTURAL DESIGN AND TRAINING

### B.1  ROI VOLUME MODELS

For each ROI Volume biomarker, we build a separate deep kernel regression model with adaptive shrinkage. The deep kernel models (p-DKGP and ss-DKGP) take as input 145 volumetric ROIs along with the following covariates: Age at Baseline, Sex, Diagnosis at Baseline, APOE4 Alleles, Education Years, and Time. The transformation function $\Phi$ is implemented as a multilayer perceptron (MLP) composed of a sequence of linear layers. $\Phi$ reduces the input dimensionality from 151

(145 imaging features + 5 covariates and Time) to 64. Based on empirical validation, further reduction degrades predictive performance. The Gaussian Process (GP) is initialized with a zero mean function and an RBF kernel.

The p-DKGP is trained for 500 epochs with a learning rate of 0.01 using the Adam optimizer (Kingma & Ba, 2014) (with a weight decay of 0.01) and a dropout rate of 0.2 for regularization. Upon completion, we save the weights $(\mathbf{W}_p, \mathbf{b}_p)$ and the GP hyperparameters (variance and lengthscale) for inference on new test subjects and for transfer learning in the subject-specific model (ss-DKGP). For the subject-specific model, we initialize the ss-DKGP with the saved weights $(\mathbf{W}_p, \mathbf{b}_p)$ and the hyperparameters of the population GP. Then, we train the ss-DKGP for 100 epochs with a learning rate of 0.01, during which the deep kernel is frozen; only the subject-specific GP hyperparameters are updated. The Adam optimizer with a weight decay of 0.05 is used in this stage.

### B.2 SPARE MODELS

For each SPARE biomarker, SPARE-AD and SPARE-BA, we build a separate deep kernel regression model with adaptive shrinkage estimation. The input features include the same 145 volumetric ROIs, along with the following covariates: Age at Baseline, Sex, Diagnosis at Baseline, APOE4 Alleles, Education, SPARE-BA, and SPARE-AD at baseline, in addition to Time.

As in the ROI Volume models, the transformation function $\Phi$ is a multilayer perceptron that projects the 153-dimensional input to a 64-dimensional feature space. We employ a GP with a zero mean function and an RBF kernel. The p-DKGP is trained for 500 epochs with a learning rate of 0.01, using the Adam optimizer with a weight decay of 0.01 and a dropout rate of 0.2. The learned weights $(\mathbf{W}_p, \mathbf{b}_p)$ and GP hyperparameters (variance and lengthscale) are then saved for subsequent inference and for initializing transfer learning in the subject-specific model. Transfer learning is performed by initializing the ss-DKGP with the saved weights $(\mathbf{W}_p, \mathbf{b}_p)$ and the population GP hyperparameters. The ss-DKGP is then trained for 100 epochs with a learning rate of 0.01, during which the deep kernel is detached from the optimization process and only the subject-specific GP hyperparameters are updated using the Adam optimizer with a weight decay of 0.05.

### B.3 DETAILS ON THE COMPETING BASELINES

We compare our method against various baselines, including Linear Mixed Effects (LMM) models, Generalized Additive Models (GAMs), Deep Regression, and the Deep Mixed Effects (DME) (Chung et al., 2019). Each baseline model is trained on the cohort of 1800 subjects, since for the development of the baselines we do not need to reserve validation-set subjects as we do for the development of the Adaptive Shrinkage Estimator. The test set is the 440 subjects. For every subject $i$, we define $U_i = (X_i, M_i, T_i)$, where $X_i$ denotes the 145 ROI Volume measurements acquired at the first visit, $M_i$ comprises the clinical covariates (age at first visit, sex, diagnosis at first visit, education years, and APOE4 alleles), and $T_i$ represents the time elapsed since the first visit. Specifically, for LMM, we use the 145 ROI Volume measurements at first visit, clinical covariates (age at first visit, sex, diagnosis at first visit, education years, APOE4 alleles) and Time as fixed effects. The Subject ID served as a random intercept and the interaction term Time:Subject ID as a slope. For GAMs, personalization involved fitting a GAM to population data of 1800 subjects, supplemented with each test subject's partially observed trajectory. The second non-linear baseline is the Deep Regression. At first, we train the Deep Regression on the population dataset of 1800 subjects. Then on the personalization, we freeze the first layers of the deep network and we fine tune only the last layer with the subject data. The architecture of the Deep Network is an MLP that consists of an input layer, three hidden layers, and an output layer. The first hidden layer contains 100 neurons, the second hidden layer has 50 neurons, and the third hidden layer again contains 100 neurons. Each hidden layer uses the Rectified Linear Unit (ReLU) activation function, which introduces non-linearity into the model and helps it learn complex data patterns. The MLP is trained using the Stochastic Gradient Descent (SGD) optimization algorithm to minimize the Mean Squared Error (MSE) loss function. For the Deep Mixed Effects (Chung et al., 2019), we used the publicly available code in order to apply the DME method to our data. As a warping mean function, we use a MLP. Additionally, we experimented with a Transformer model (Vaswani et al., 2017) utilizing positional encoding along the temporal dimension and implemented LSTM models Hochreiter & Schmidhuber (1997). However, both models faced convergence issues during training and did not

yield satisfactory results on our sparse temporal dataset. Theoretically, Transformer models rely on self-attention mechanisms to capture dependencies across sequences, which assume the availability of comprehensive and densely sampled sequential data. In the context of sparse temporal data, the self-attention mechanism cannot function optimally due to insufficient temporal information, leading to suboptimal performance. Similarly, LSTM models require temporally aligned and regularly sampled data to maintain the sequential relationships inherent in time series. Without prior preprocessing, such as data imputation to handle irregularities and missing values, LSTMs struggle to learn effectively from sparse temporal data. As a result, we omitted these models from the quantitative comparisons in the current work.

## C  ANALYSIS ON POSTERIOR CORRECTION

Our goal is to determine the oracle shrinkage parameter $\alpha$ in Equation equation 11, which combines the predictions from the population model (p-DKGP) and the subject-specific model (ss-DKGP). To achieve this, we propose minimizing the Mean Squared Error (MSE) between the combined prediction $y_c$ and the ground truth $y_t$ over all time points. The objective function is defined as:

$$J(\alpha) = \sum_{t=0}^{t_n} \left(y_t - (\alpha y_{p_t} + (1 - \alpha)y_{s_t})\right)^2. \tag{11}$$

In this section, we provide a theoretical justification for this formulation, explaining why the independence assumption between the models' errors does not affect the estimation of $\alpha$ using this objective function.

Both the p-DKGP and ss-DKGP models provide predictive means $y_{p_t}$ and $y_{s_t}$ for the ROI value at each time point $t$. We aim to find the oracle $\alpha$ that minimizes the MSE between the combined prediction $y_c$ and the ground truth $y_t$. The combined prediction is given by:

$$y_c = \alpha y_{p_t} + (1 - \alpha)y_{s_t}. \tag{12}$$

To find the optimal $\alpha$, we take the derivative of $J(\alpha)$ with respect to $\alpha$ and set it to zero:

$$\frac{dJ}{d\alpha} = -2 \sum_{t=0}^{t_n} \left(y_t - (\alpha y_{p_t} + (1 - \alpha)y_{s_t})\right)(y_{p_t} - y_{s_t}) = 0. \tag{13}$$

Simplifying, we get:

$$\sum_{t=0}^{t_n} \left(y_t - (\alpha y_{p_t} + (1 - \alpha)y_{s_t})\right)(y_{p_t} - y_{s_t}) = 0. \tag{14}$$

Solving for $\alpha$, we find:

$$\alpha^* = \frac{\sum_{t=0}^{t_n}(y_t - y_{s_t})(y_{p_t} - y_{s_t})}{\sum_{t=0}^{t_n}(y_{p_t} - y_{s_t})^2}. \tag{15}$$

This expression shows that the optimal $\alpha$ depends on the covariance between $y_t - y_{s_t}$ and $y_{p_t} - y_{s_t}$, and the variance of $y_{p_t} - y_{s_t}$.

To gain further insight into the dependence of the optimal $\alpha^*$ on statistical properties of the data, we relate Equation 15 to the concepts of covariance and variance. Let us define:

$$X_t = y_{p_t} - y_{s_t}, \quad Y_t = y_t - y_{s_t}. \tag{16}$$

With these definitions, Equation 15 becomes:

$$\alpha^* = \frac{\sum_{t=0}^{t_n} Y_t X_t}{\sum_{t=0}^{t_n} X_t^2}. \tag{17}$$

The numerator and denominator in Equation 17 are related to the sample covariance and variance, respectively. Specifically, the numerator is proportional to the covariance between $Y_t$ and $X_t$, and

the denominator is proportional to the variance of $X_t$:

$$\text{Cov}(Y, X) = \frac{1}{n} \sum_{t=0}^{t_n} (Y_t - \bar{Y})(X_t - \bar{X}), \tag{18}$$

$$\text{Var}(X) = \frac{1}{n} \sum_{t=0}^{t_n} (X_t - \bar{X})^2, \tag{19}$$

where $\bar{Y}$ and $\bar{X}$ are the sample means of $Y_t$ and $X_t$, respectively, and $n = t_n + 1$ is the number of time points.

Assuming that $Y_t$ and $X_t$ are centered (i.e., $\bar{Y} = 0$ and $\bar{X} = 0$), which is valid if we consider deviations from their means, Equation 17 simplifies to:

$$\alpha^* = \frac{n \cdot \text{Cov}(Y, X)}{n \cdot \text{Var}(X)} = \frac{\text{Cov}(Y, X)}{\text{Var}(X)}. \tag{20}$$

This expression shows that the optimal $\alpha^*$ is the coefficient that minimizes the residual sum of squares in a simple linear regression of $Y_t$ on $X_t$ without an intercept. In other words, $\alpha^*$ is the scaling factor that best relates the difference between the population and subject-specific predictions ($X_t$) to the residuals of the subject-specific model ($Y_t$).

- If $\text{Cov}(Y, X)$ is large and positive, it indicates that when the subject-specific model underpredicts or overpredicts ($Y_t$ deviates from zero), the difference between the population and subject-specific predictions ($X_t$) tends to be in the same direction. In this case, a larger $\alpha$ (giving more weight to the population model) helps reduce the overall error.

- If $\text{Cov}(Y, X)$ is small or negative, it suggests that the population model does not provide useful information to correct the subject-specific model's errors, and a smaller $\alpha$ (giving more weight to the subject-specific model) is preferable.

This analysis confirms that the optimal $\alpha^*$ depends on the covariance between $y_t - y_{s_t}$ and $y_{p_t} - y_{s_t}$, and the variance of $y_{p_t} - y_{s_t}$. Understanding this dependence provides valuable insight into how the differences between the models' predictions relate to the residuals and how to optimally combine them to minimize the prediction error.

### C.1 INDEPENDENCE ASSUMPTION AND ITS IMPACT

The combined predictive mean $y_c$ is a deterministic function of $y_{p_t}$, $y_{s_t}$, and $\alpha$, as given in Equation 12. It does not involve the errors or variances associated with the predictions. As a result, the independence or correlation between the models' errors does not influence the calculation of $y_c$. While the independence assumption does not affect the estimation of $\alpha$ or the calculation of $y_c$, it does impact the calculation of the combined predictive variance $v_c$. The variance of the combined prediction is given by:

$$v_c = \alpha^2 v_{p_t} + (1 - \alpha)^2 v_{s_t} + 2\alpha(1 - \alpha)\text{Cov}(y_{p_t}, y_{s_t}). \tag{21}$$

If the errors of the two models are assumed to be independent, the covariance term $\text{Cov}(y_{p_t}, y_{s_t})$ is zero, simplifying $v_c$ to:

$$v_c = \alpha^2 v_{p_t} + (1 - \alpha)^2 v_{s_t}. \tag{22}$$

Empirical analysis indicates that the errors of the two models are mildly correlated, with correlation to range between $0.136$ to $0.394$. Therefore, the inclusion the covariance term in the calculation of $v_c$ to accurately quantify the uncertainty of the combined prediction.

Overall, the theoretical justification demonstrates that the MSE objective function is appropriate for estimating the shrinkage parameter $\alpha$ in our context. It avoids the need for the independence assumption during $\alpha$ estimation and simplifies the optimization process. However, when calculating the predictive variance $v_c$, it is essential to account for the covariance between the models' predictions to accurately quantify uncertainty.

To address this issue, we do:

- **Estimating Covariance:** Empirically estimate $\text{Cov}(y_{p_t}, y_{s_t})$ using validation data.

Table 3: Correlation between the errors of p-DKGP and ss-DKGP models for different ROI Volume biomarkers and Observations

| ROI | 3 Observations | 4 Observations | 5 Observations | 6 Observations |
|---|---|---|---|---|
| Hippocampus R | 0.237 | 0.337 | 0.374 | 0.318 |
| Thalamus Proper R | 0.136 | 0.348 | 0.302 | 0.344 |
| Lateral Ventricle R | 0.201 | 0.247 | 0.319 | 0.354 |
| Hippocampus L | 0.341 | 0.300 | 0.348 | 0.208 |
| Amygdala R | 0.262 | 0.325 | 0.355 | 0.372 |
| Amygdala L | 0.292 | 0.356 | 0.331 | 0.394 |

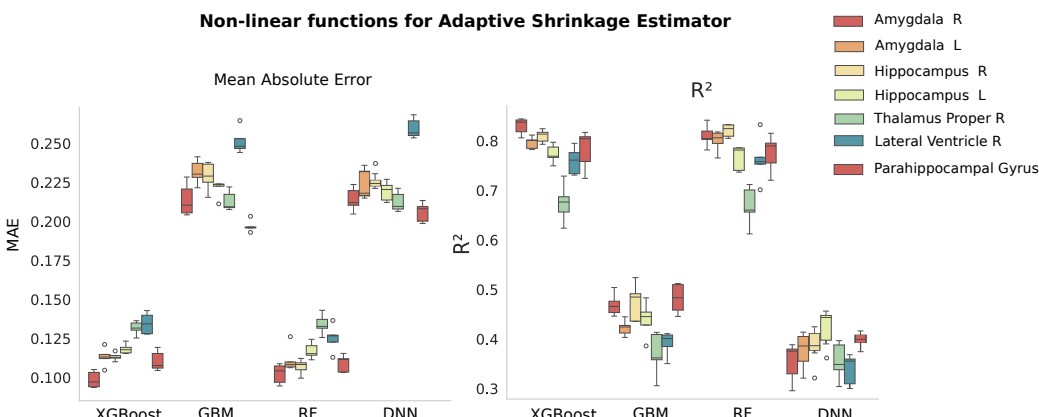

Figure 5: We present MAE and $R^2$ from 5-fold cross-validation using the 200 held-out subjects from ADNI and BLSA subjects for the Adaprive Shrinkage estimator using XGBoost, GBM, RF and DNN as non-linear functions

- **Adjusting Variance Calculations:** Include the covariance term in the calculation of $v_c$ as per Equation 21.

- **Reassessing Prediction Intervals:** Recompute prediction intervals using the adjusted $v_c$ to ensure improved coverage.

## C.2 ALTERNATIVES OF NON-LINEAR FUNCTIONS FOR ADAPTIVE SHRINKAGE ESTIMATOR

We experiment with several non-linear functions to determine which one learns best the adaptive shrinkage mapping, namely the mapping between $a$ and $y_p, y_s, V_p, V_s, T_{obs}$. We conduct 5-fold cross-validation using XGBoost Regression (XGBoost), Random Forest (RF), Gradient Boosting Machine (GBM), and a Deep Neural Network (DNN). The DNN architecture includes a linear layer (5x16), ReLU activation, a linear layer (16x8), ReLU activation, and a final linear layer (8x1). It is trained with MSE loss and optimized using Adam with a learning rate of 0.01. Results, presented in Figure 5 indicate that XGBoost Regression and Random Forest achieve the best performance in terms of mean absolute error and $r^2$ score on the test set, with both models achieving an average $r^2$ score greater than 0.75 across the majority of the ROI Volumes.

## D EXPERIMENTS

### D.1 STRATIFIED PERFORMANCE ANALYSIS BY COVARIATES

To thoroughly evaluate our method, we perform stratification of prediction errors across key demographic and clinical factors: sex, APOE4 Allele status, and education level. This stratification allows us to examine the model's ability on varying subpopulations. We report the Mean Absolute Error and corresponding 95% confidence intervals (CIs) for pers-DKGP, alongside with the competing baselines.

Table 4: XGBoost performance on predicting the adaptive shrinkage $\alpha$ for 7 ROI Volume biomarkers

| ROI Volume | MAE | R² |
|---|---|---|
| Amygdala R | 0.099 | 0.830 |
| Amygdala L | 0.113 | 0.796 |
| Hippocampus R | 0.113 | 0.810 |
| Hippocampus L | 0.118 | 0.774 |
| Lateral Ventricle R | 0.132 | 0.675 |
| Thalamus Proper R | 0.135 | 0.759 |
| PHG R | 0.111 | 0.783 |

**Stratification by Sex.** Our results indicate that pers-DKGP consistently achieves the lowest Mean AE for both male and female groups. In males, pers-DKGP attains a Mean AE of $0.135$ (95% CI: $[0.120, 0.150]$), significantly outperforming LMM, which yields a Mean AE of $0.187$ (CI: $[0.160, 0.214]$). Similarly, for females, pers-DKGP reports a Mean AE of $0.145$ (CI: $[0.130, 0.160]$), compared to GAM's Mean AE of $0.198$ (CI: $[0.165, 0.231]$). Although prediction errors are slightly higher in females—likely due to increased biomarker variability—the consistently narrower CIs of pers-DKGP underscore its enhanced reliability across sexes.

**Stratification by APOE4 Alleles Status.** Considering the crucial role of the APOE4 Allele in Alzheimer's Disease progression, we examine model performance for Non-Carriers, Heterozygous and Homozygous separately. For APOE4 homozygotes, pers-DKGP achieves a Mean AE of $0.142$ (CI: $[0.128, 0.156]$), markedly lower than DME's Mean AE of $0.210$ (CI: $[0.176, 0.244]$). For non-carriers, pers-DKGP obtains a Mean AE of $0.130$ (CI: $[0.118, 0.142]$), outperforming DeepRegr, which records a Mean AE of $0.192$ (CI: $[0.162, 0.222]$).

**Stratification by Education** Education level, serving as a proxy for cognitive reserve, introduces additional variability in disease progression predictions. In the subgroup with education levels below 16 years, pers-DKGP achieves a mean AE of $0.155$ (CI: $[0.140, 0.170]$), outperforming LMM, which exhibits a mean AE of $0.225$ (CI: $[0.195, 0.255]$). Among subjects with 16 or more years of education, pers-DKGP maintains its advantage, recording a mean AE of $0.120$ (CI: $[0.110, 0.130]$), whereas GAM shows a mean AE of $0.175$ (CI: $[0.145, 0.205]$).

Overall, the stratification of AE demonstrates that pers-DKGP outperforms baseline methods in all subpopulations. Its lower mean AE and narrower confidence intervals indicate not only higher predictive accuracy but also greater reliability, even in challenging subgroups such as APOE4 carriers, and individuals with lower education levels.

### D.2 PERFORMANCE WITH NUMBER OF OBSERVATIONS

**Error with Number of Observations for SPARE-AD Score.** Table 5 presents the mean absolute error and 95% confidence interval for the SPARE-AD biomarker across different numbers of observations (history). A history of 1 corresponds to using the population model prediction, which we employ when only a single acquisition of the subject is available; in this case, we have $\alpha = 1$. As we increase the number of observations, we apply posterior correction with adaptive shrinkage $\alpha$ inferred by the adaptive shrinkage estimator, allowing us to adjust the model based on the subject's individual history. Notably, the mean AE decreases as more observations are included. This demonstrates the benefit of applying Adaptive Shrinkage with increased subject history to improve the accuracy of the SPARE-AD biomarker prediction.

Table 5: Mean Absolute Error and 95% Confidence Interval for the SPARE-AD biomarker with increasing number of observations

| Observations | Mean AE | 95% CI |
|---|---|---|
| 1 ($\alpha = 1$) | 0.227 | 0.003 |
| 2 | 0.233 | 0.008 |
| 3 | 0.219 | 0.008 |
| 4 | 0.153 | 0.010 |
| 5 | 0.148 | 0.010 |

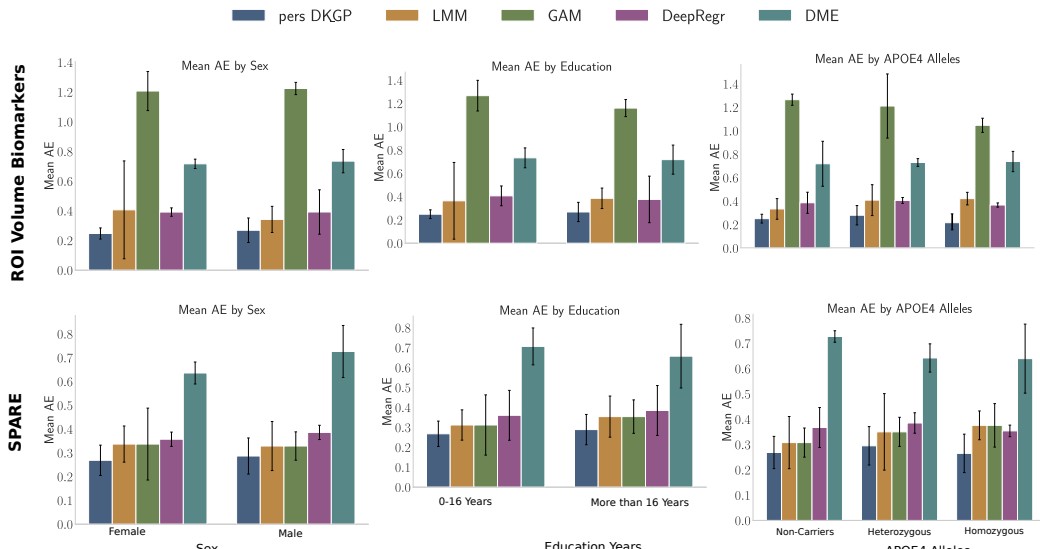

Figure 6: We stratify MAE by key covariates—Sex, APOE4 Alleles, and Education Years—to rigorously assess model performance across different subpopulations. Error bars denote the 95% confidence intervals of the MAE. The top row aggregates metrics for seven ROI Volume biomarkers, while the bottom row summarizes the MAE for both SPARE-AD and SPARE-BA.

**Error Analysis.** Figure 7**a** illustrates the distribution of absolute errors across history levels (1 to 6) using boxplots. The median error is indicated by the central line within each box, with the interquartile range (IQR) defining the edges and whiskers extending to 1.5 times the IQR. Outliers are depicted as individual points beyond the whiskers. A red line represents the mean absolute error, providing an overview of the central tendency.

The results demonstrate a marked reduction in mean absolute error with increasing history, particularly during the earlier transitions: a 21.96% decrease from history 1 to 2 and a further 15.92% decrease from history 2 to 3. This underscores the significance of incorporating additional longitudinal observations. However, the improvements plateau at higher history levels, reflecting diminishing returns. It is important to note that the error will never practically reach zero, owing to the inherent noise and variability of neuroimaging biomarkers. Nevertheless, the results highlight the necessity of subject-specific personalization, as individual trajectories often deviate from population-level SPARE-AD estimates. With additional follow-up observations, these deviations are better captured, resulting in more accurate and individualized SPARE-AD trajectories. This emphasizes the critical role of model adaptation in clinical practice, as refined SPARE-AD estimates can provide valuable insights for predicting disease progression, including transitions to dementia or, more specifically, progression from MCI to Alzheimer's Disease.

### D.3    QUALITATIVE EXAMPLES OF ROI VOLUME AND SPARE BIOMARKERS

In this section we provide additional qualitative results on test subjects. We present results for the ROI volume biomarkers as well as the SPARE AD biomarker. The ROI progression models use as input the imaging scan (145 Volumetric ROIs), demographics and clinical variables. The SPARE-AD progression model uses the 145 Volumetric ROIs, demographics and clinical variables as well as the SPARE-AD score at baseline.

**Empirical Evidence of Predicted SPARE-AD Trajectories for MCI Progressor.** In figure 8 we present an example of a subject that starts as Cognitive Normal at the Age of 74 years old. We use our model (pers-DKGP) in order to predict the longitudinal SPARE-AD changes from the 145 Volumetric ROIs as well as the demographics (Age, Sex, Education Years) and clinical variables such as the Clinical Diagnosis and the APOE4 Alleles. At the first visit of the subject, we extrapolate a SPARE-AD trajectory that indicates no changes related to progression. Within the 2 and a half years of observations the MCI the predicted trajectory of the SPARE-AD biomarker indicates no significant longitudinal change in the SPARE-AD trajectory. In the 42 months of observations,

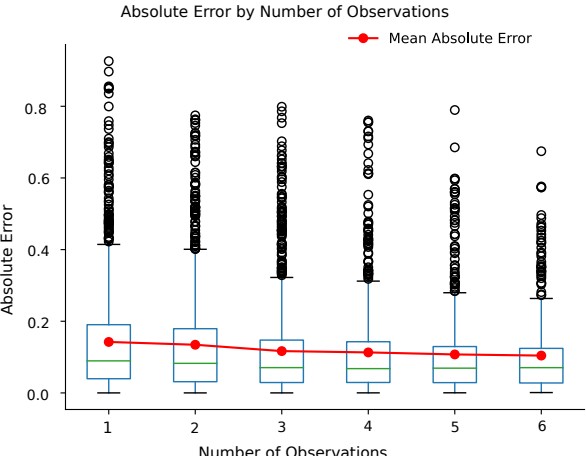

Figure 7: Boxplots show the distribution of absolute errors across history levels (1 to 6), with the central line indicating the median, the box edges representing the interquartile range (IQR), and whiskers extending to 1.5 times the IQR. Outliers are shown as points beyond the whiskers. The red line connects the mean absolute error for each level.

the predicted SPARE-AD trajectory indicates an increasing trend in the SPARE-AD values that indicates increased AD released patterns in the brain. Increased AD-like patterns indicate higher risk of conversion to MCI or Dementia (AD). In almost 5 years of observation, the predicted trajectory indicates a steeper increase in the future SPARE-AD values indicating againg high risk of MCI or AD. The subject finaly is clinically diagnosed with MCI after 80 months of observation. Our method is able to predicted changes of biomarker values that are indicative of Progression and this highlights also the clinical usage of our method as a stong predictive tool for progression prediction either for use in the clinical practice or the design of clinical trials. For example, this subject with an increasing trend of SPARE-AD trajectory would be an ideal subject for recruitment in a clinical trial as it converts to demonstrates inclining biomarker trajectory making it a subject that is highly likely to be part of a clinical trial.

Figure 8: We present predicted SPARE-AD trajectories for a Cognitive Normal subject at the baseline age of 74 years old. After 7 years the subject is diagnosed with Mild Cognitive Impairment. The predicted SPARE-AD trajectories predict the increasing attrophy-like patterns 3 years prior the clinical diagnosis of conversion to MCI. This highlights the potential clinical application of our tool for progression prediction and clinical trial design.

**Empirical Evidence from a Healthy Control and and MCI Progressor.** In Figure 9, we present a qualitative comparison of predicted trajectories for two subjects who begin the study at similar ages—74 (left) and 71 (right), respectively—and are cognitively normal at baseline. We analyze the volumetric loss in three brain regions: the amygdala, hippocampus, and lateral ventricle. The volumetric loss is modeled as a function of MRI scans alongside clinical and demographic covariates, including age, sex, diagnosis, APOE4 allele status, and years of education.

At the initial visit, both subjects exhibit minimal hippocampal atrophy. However, over successive follow-up observations, the subject on the left ( 9b) demonstrates a markedly steeper decline in hippocampal volume compared to the subject on the right, who maintains a more stable hippocampal trajectory. The predicted accelerated decline in hippocampal volume for the subject on the left suggests an elevated risk of progressing to mild cognitive impairment (MCI) or dementia, potentially due to underlying pathology such as Alzheimer's disease (AD) or accelerated brain aging. In contrast, the subject on the right ( 9a), who remains a healthy control throughout the observation period, exhibits only minimal hippocampal volume loss.

This example illustrates the practical application of our method in predicting disease progression, which has significant implications for clinical practice, clinical trial design, and treatment effect estimation. Specifically, in the context of clinical trial design, identifying subjects with steep hippocampal atrophy trajectories can inform the recruitment of individuals who are more likely to exhibit disease progression, thereby enhancing the efficiency and efficacy of the study.

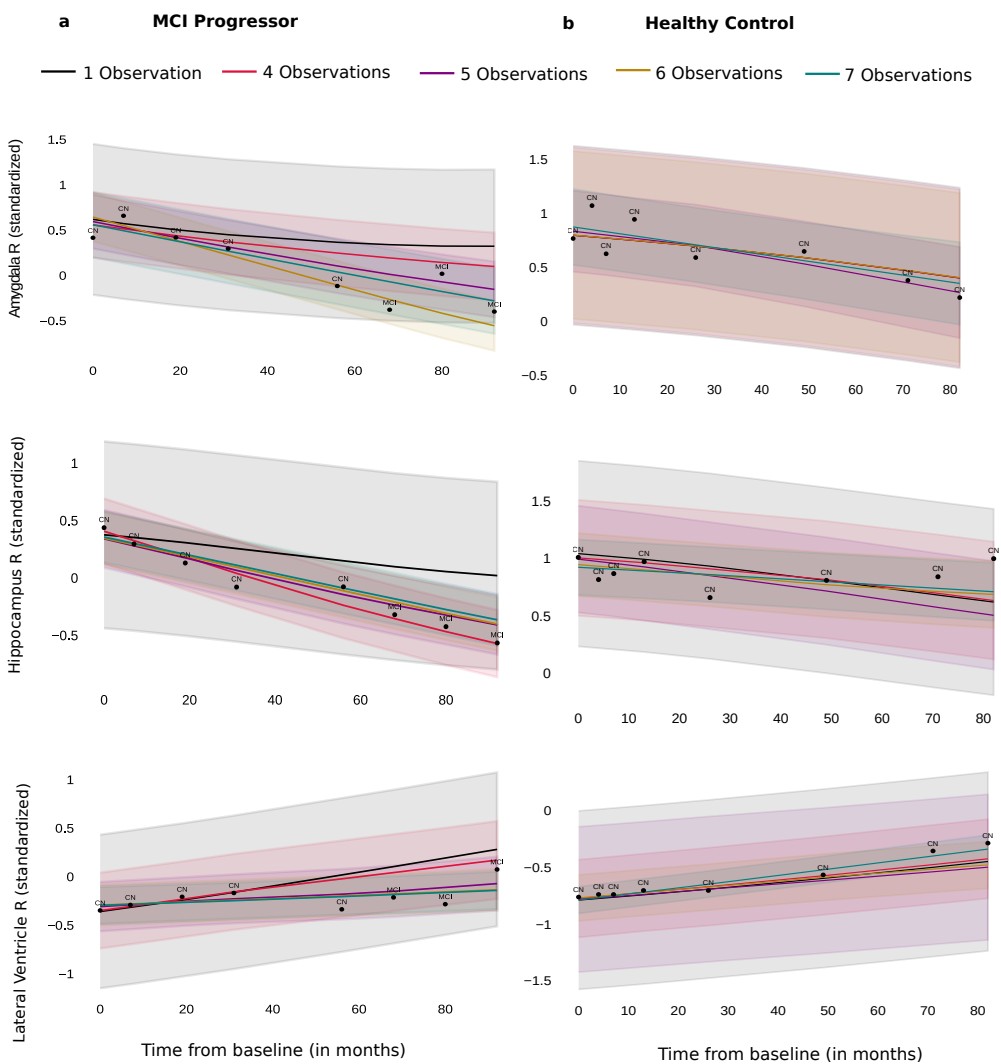

Figure 9: We present predicted Amygdala and Hippocampal Volume trajectories and Ventricular Enlargement for a Healthy Control and and MCI Progressor. MCI Progressor exhibits steeper volume loss in Amygdala and Hippocampus in comparison with the Healthy Control. MCI Progressor exhibits either accelarated brain aging or is in the onset of AD which justifies its faster volume loss.

**Empirical Evidence of the Personalization in Test Subjects.** To further validate the efficacy of our method, we provide empirical evidence through qualitative analysis in scenarios where individual trajectories either diverge from or align with the true underlying trend. In Figure 10, we present a cohort of test subjects (panels **(a)**–**(h)**) exhibiting variability in progression status, alongside the corresponding adaptive shrinkage parameter $\alpha$—depicted in the second row—utilized at each personalization step. Consistently across all examples, we observe that the adaptive shrinkage parameter $\alpha$ progressively decreases as the number of observations increases. In several cases, the adjustments remain more conservative, with $\alpha$ staying closer to 1, which aligns with the foundational intuition of our method. This pattern suggests that an adequate accumulation of evidence regarding a subject's trajectory is necessary to shift the adaptive shrinkage parameter toward zero, thereby placing greater trust in the ss-DKGP predictions. This rationale is well-founded, as substantial evidence is crucial for the ss-DKGP to generate meaningful trajectories and mitigate the noise variations inherent in neuroimaging data acquisitions. Additional examples are visualized in Figure 11.

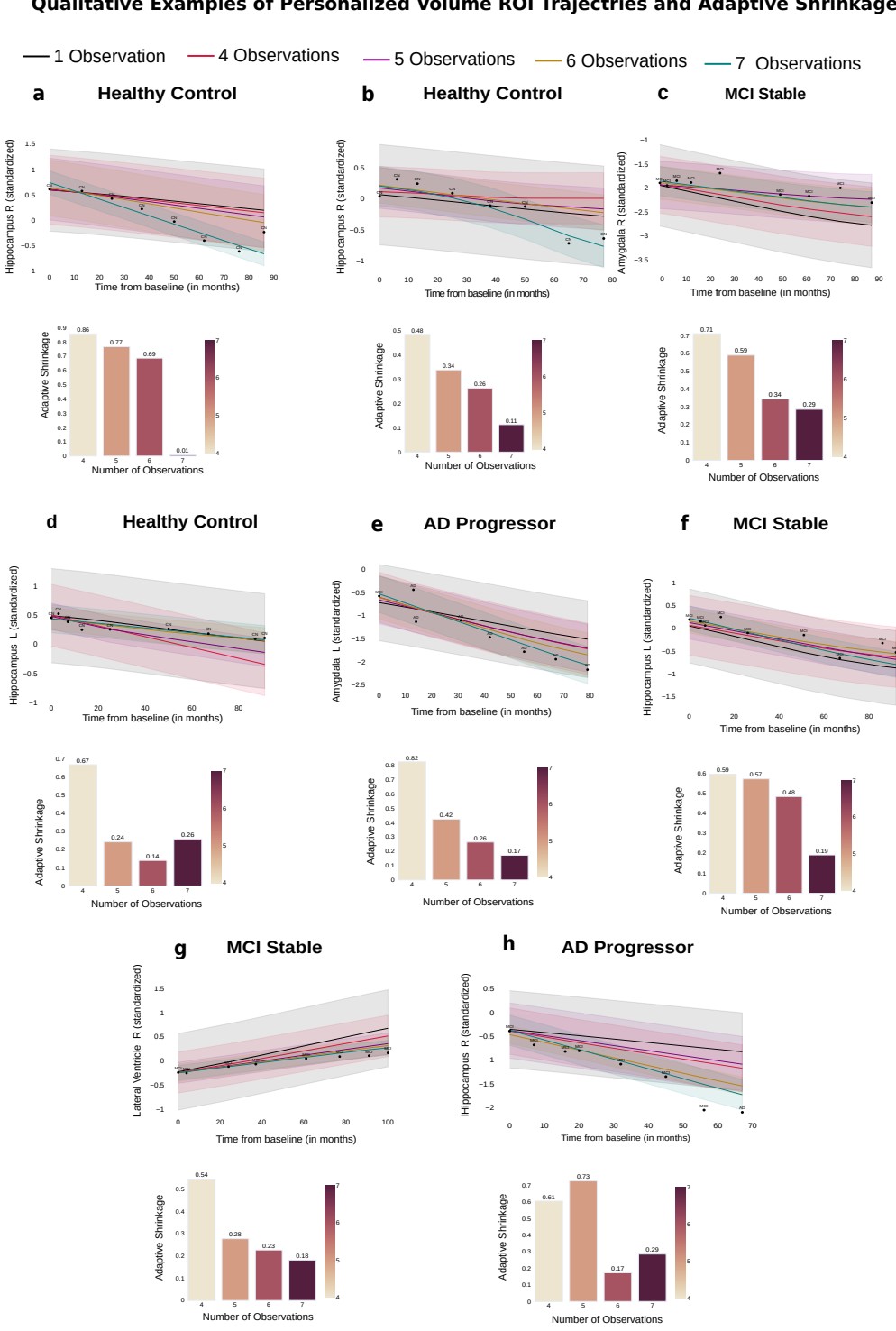

Figure 10: We present qualitative examples where population trajectories deviate from the subject's observed trajectory throughout the observation period (in years). Evidence is provided from eight distinct test subjects. In the first row of each panel (**a**)-(**h**), we present the adapted trajectories. The second row of each subfigure visualizes the corresponding adaptive shrinkage for posterior correction for each observation, ranging from 4 to 7 observations.

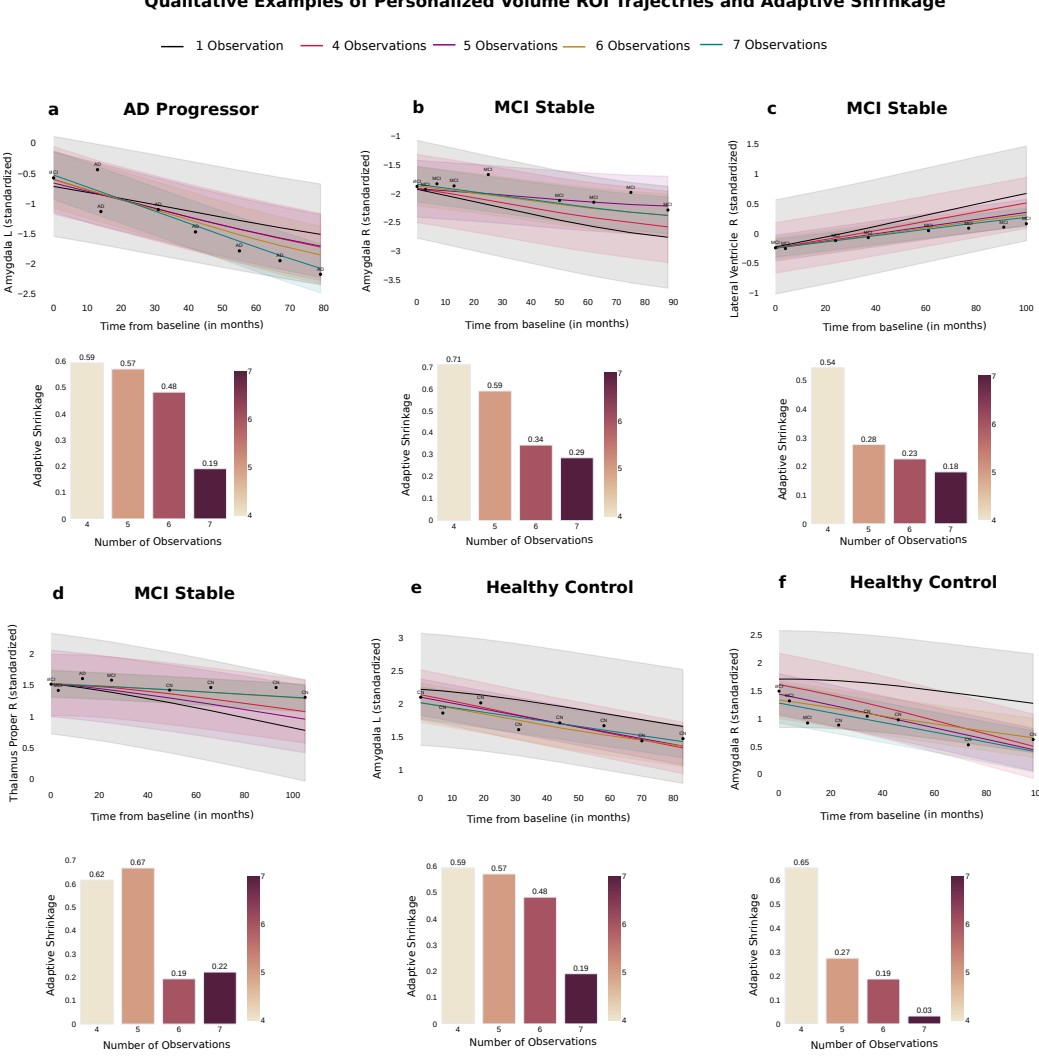

Figure 11: We present qualitative examples where population trajectories deviate from the subject's observed trajectory throughout the observation period (in years). Evidence is provided from six distinct test subjects. In the first row of each panel (**a**)-(**f**), we present the adapted trajectories. The second row of each subfigure visualizes the corresponding adaptive shrinkage for posterior correction for each observation, ranging from 4 to 7 observations.

## D.4 ANALYSIS OF ADAPTIVE SHRINKAGE ESTIMATOR

### D.4.1 ABLATION ON SHRINKAGE PARAMETER $\alpha$

Determining the shrinkage for each ROI Volume is non-trivial, particularly for predicting long-term trajectories. This is a difficult task because either a subject's trajectory would deviate from population trends, or a subject would have limited acquisitions, making it difficult for the subject-specific model to extrapolate its ROI Volume trajectory. Volume loss in the brain is a slow process, especially for a subject who is young or has not yet developed any pathology. Thus, in the case of limited acquisitions for a subject, which are also close in time to the baseline, the additional observations are rather noisy copies of the baselines and do not contain any "signal" of the trajectory of developing atrophy. In that case, the ss-DKGP model would not have enough evidence to extrapolate future ROI Volume. As a result, we should find the ideal shrinkage to combine the two predictors and eventually leverage both the population's ability to make reliable long-term predictions and the subject-specific model's ability to learn short-term predictions. We show that adaptive shrinkage provides the best results compared to any other weighting scheme, as we also present in table 6.

Table 6: Ablation study on the shrinkage parameter $\alpha$. We report the Mean AE along with its 95% percentile CI, Mean Coverage, and Mean Interval Width

| ROI | Mean AE (CI) | Mean Coverage | Mean Interval |
|---|---|---|---|
| **Best Constant** | | | |
| Hippocampus R | 0.257 (0.209) | 0.808 | 0.843 |
| Lateral Ventricle R | 0.143 (0.182) | 0.853 | 0.507 |
| Thalamus Proper R | 0.241 (0.214) | 0.934 | 1.127 |
| Amygdala R | 0.349 (0.317) | 0.742 | 0.918 |
| Hippocampus L | 0.274 (0.245) | 0.805 | 0.850 |
| PHG R | 0.423 (0.360) | 0.582 | 0.844 |
| **Deterministic** | | | |
| Hippocampus R | 0.308 (0.275) | 0.480 | 0.459 |
| Lateral Ventricle R | 0.156 (0.192) | 0.620 | 0.310 |
| Thalamus Proper R | 0.308 (0.287) | 0.512 | 0.492 |
| Amygdala R | 0.418 (0.400) | 0.503 | 0.650 |
| Hippocampus L | 0.314 (0.290) | 0.487 | 0.478 |
| PHG R | 0.487 (0.457) | 0.459 | 0.681 |
| **Adaptive Shrinkage** | | | |
| Hippocampus R | **0.243** (0.191) | 0.795 | 0.902 |
| Lateral Ventricle R | **0.131** (0.186) | 0.855 | 0.626 |
| Thalamus Proper R | **0.219** (0.216) | 0.849 | 0.911 |
| Amygdala R | **0.312** (0.283) | 0.762 | 0.964 |
| Hippocampus L | **0.258** (0.241) | 0.790 | 0.901 |
| PHG R | **0.389** (0.344) | 0.745 | 0.908 |

### D.4.2 INTERPRETATION OF ADAPTIVE SHRINKAGE ESTIMATOR

As we increase the number of observations, we see that, no matter the biomarker, the alpha tends to zero. This aligns with the domain expectation that the longer the time from the baseline of the last observation Tobs, the more likely we are to have observed a trajectory trend from the subject's data.

In Figure 12, we visualize the distribution of adaptive shrinkage in the test set as well as in the three external clinical studies. This demonstrates that adaptive shrinkage has learned to assign greater trust to the subject-specific model as the number of follow-ups increases for a subject. This aligns perfectly with domain expectations and the explainability analysis we implemented for the Adaptive Shrinkage Estimator. This property makes the Adaptive Shrinkage Estimator a transparent method for performing posterior correction in the two predictive distributions, p-DKGP and ss-DKGP.

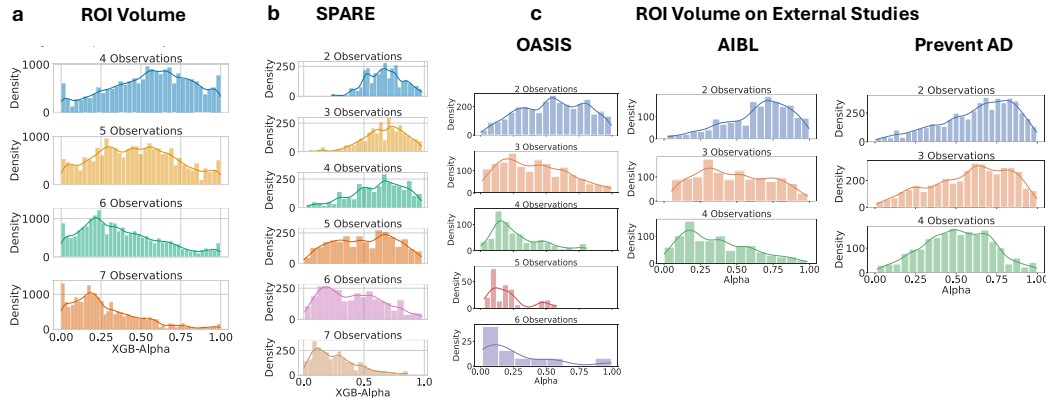

Figure 12: We visualize the distribution of adaptive shrinkage $\alpha$ for **a)** the 7 ROI Volumes, the **b)** SPARE-BA and SPARE-AD biomarkes and **c)** the 7 ROI Volumes in the external neuroimaging studies: OASIS, AIBL and PreventAD

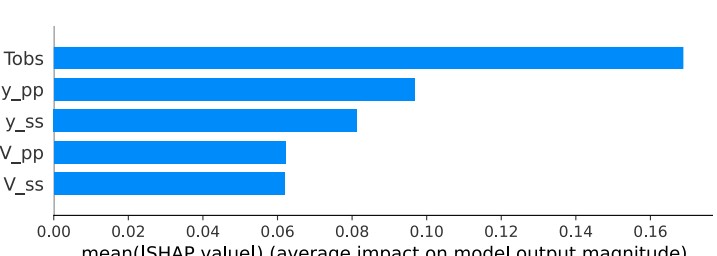

Figure 13: We calculate SHAP values for the Adaptive Shrinkage Estimator for the SPARE-AD biomarker. As expected, the time of observation Tobs emerges as the most influential feature of the Adaptive Shrinkage stimator.

Table 7: Correlation Analysis between Deviation ($\delta_y$) and Predicted $\alpha$, and between $T_{\text{obs}}$ and Predicted $\alpha$ for Large Deviation

| Biomarker | Correlation between $T_{\text{obs}}$ and Predicted $\alpha$ for Large $\delta_y$ |
|---|---|
| SPARE-BA | -0.640 |
| SPARE-AD | -0.529 |
| Lateral Ventricle | -0.484 |
| Hippocampus L | -0.401 |
| Hippocampus R | -0.381 |
| Thalamus Proper R | -0.555 |
| PHG R | -0.479 |
| Amygdala R | -0.439 |

### D.5 COMPARISON ON ALTERNATIVE GP PERSONALIZATION APPROACHES

In this section, we conduct a comparative analysis with other personalized GPs that align with our formulation. Specifically, within the ss-DKGP framework, we do an ablation study to see how the $\Phi$ transformation, learned from the population model, affects the subject-specific process. To achieve this, we train the ss-DKGP for each subject on the test set without initializing the deep kernel (ss-DKGP no init). We also train a standard subject-specific Gaussian Process (ss-GP) with a zero mean and RBF kernel. This comparison demonstrates the effectiveness of transferring the $\Phi$ from the p-DKGP when training the ss-DKGP. Additionally, we explore an alternative personalization approach where the population dataset $D_p$ is augmented with the subject's observed trajectory $D_s$. In this setting, we again employ the $\Phi$ transformation learned from the p-DKGP. This approach is referred to as the ft-DKGP (fine-tuned DKGP). We perform transfer learning by initializing the weights of the deep kernel with $(\mathbf{W}_p, \mathbf{b}_p)$. The ft-DKGP is trained for 500 epochs using the same learning rate as the p-DKGP. During this process, the deep kernel is detached from the optimization procedure, and only the hyperparameters of the subject-specific GP are updated. The Adam optimizer with a weight decay of 0.05 is utilized.

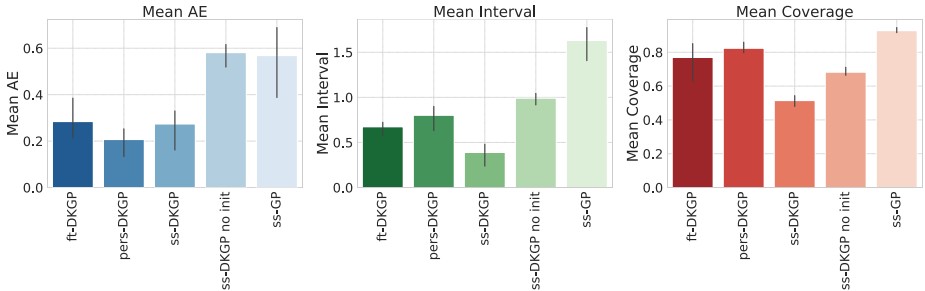

Figure 14: Comparison of predictive performance and uncertainty quantification across various GP models, averaged over three regions of interest (ROIs): Hippocampus, Lateral Ventricle, and Thalamus Proper, indicating that the personalized DKGP models achieve the best prediction accuracy and highest coverage. The bar plot displays the Mean performance metrics (AE, interval length and coverage) across these ROIs, while the line represents the standard deviation.

Among the ss-DKGP, ss-DKGP no init, and ss-GP models, we observe that ss-DKGP achieves the lowest Absolute Error (AE) with a significant margin compared to the other two settings. This indicates that leveraging the population transformation $\Phi$ is crucial for the effective training of ss-DKGP. This finding supports our hypothesis that the transformation $\Phi$ successfully captures the most predictive features for ROI progression, which are beneficial for ss-DKGP training. We observe that ft-DKGP model achieves performance that is close to the ss-DKGP model. However, ft-DKGP fails to personalize on unseen times, since the predicted trajectory falls back to the population trend. This is not the optimal way to personalize since the trajectory does not adapt to the subject specific trend. Additionally, it is not computationally efficient to retrain the model with the entire population data every time we need to personalize a subject.

Furthermore, the pers-DKGP model achieves the lowest AE, which is an additional indication in favor of our approach. It highlights the strength of including the p-DKGP model in the final personalized prediction. Knowing solely the observed trajectory of a subject is not enough in case of limited and noisy observations. In that case we should trust the p-DKGP model more, which translates to an $\alpha$ parameter close to 1. Interestingly, this intuition aligns with the predicted $\alpha$ that we got during the personalization from the XGBoost regression. To verify that, we gathered the predicted $\alpha$ from the personalization process from the 7 ROIs. We plotted the distribution of $\alpha$ with the number of observations ranging from 4 till 7. The plot is shown in Figure 12**a**. It clearly depicts that, as the number of observations increases, the distributions tend to show more noticeable skewness to the right, with higher densities in the lower $\alpha$ ranges and decreasing densities towards higher $\alpha$ values. This trend suggests that as more observations are taken into account in personalization, the shrinkage parameter $\alpha$ tends to be smaller. That translates to more trust to the ss-DKGP prediction. This is highly intuitive because as observation time $T_{obs}$ increases, more acquisitions we obtain for a subject and thus the more information the ss-DKGP captures about the progression of a ROI over time.

