# OpenReview forum: "Adaptive Shrinkage Estimation for Personalized Deep Kernel Regression in Modeling Brain Trajectories"
_ICLR.cc/2025/Conference — ICLR 2025 Poster_

### Official Review · Reviewer_enPj · 2024-10-28

**Soundness:** 3
**Presentation:** 3
**Contribution:** 2
**Rating:** 6
**Confidence:** 4

**Summary:**

The authors have developed a deep kernel regression model to predict longitudinal brain volume changes. The model is made for personalised predictions and has two parts, one population model for global prediction and one subject-specific model for personalised predictions. The model is compared to several commonly used models on two open data sets. The resulting models are further evaluated on three external test sets. The results are convincing, and the method appears to work well.

**Strengths:**

The paper is well-written and in general easy to follow. The method is straight-forward, clear, and appears to work well, and is applied with convincing results to real-world datasets. I really like that you evaluated on multiple external data; this strengthens the confidence in and validity of the results.

**Weaknesses:**

It is not entirely clear where the methodological novelty lies. Personalised GPs have been proposed before (as mentioned in the introduction) and weighting schemes have also been proposed before (also correctly mentioned in the introduction). The adaptive shrinkage is interesting, and appears to work well, but must also be considered a small contribution.

It is great that the method is evaluated on no less than three external test sets, but it would have been good/more convincing if it was also trained on more than just two training sets.

There should be en ablation for the personalisation (or, equivalently, presenting results with \alpha=1), to see if it actually improves the results over just having the population level model.

There are not confidence intervals in the bar plots, which means it is not possible to compare those results.

**Questions:**

Comments/questions/required changes:
 - How much data from individuals do you need for the personalised GP to be useful/to help in the predictions?
 - The step on Line 12 of Algorithm 1 is not atomic. Split this up or explain clearly somewhere what is being done.
 - It says the values in parentheses in Table 1 are standard deviations. Present the standard errors (standard deviations of the means) instead, so that the results can actually be compared properly.

Minor comments:
 - Do use \citet rather than \cite or \citep when you mention a reference in the body of the text, so that the names are not in parentheses in those cases.
 - Some more LaTeX-related issues: Remove blank lines after equations to avoid indented sentences after. If an equation ends a sentence, put a full stop after it (or, as in Equation 7, put the comma after the equation, not at the beginning of the next sentence). In general, make equations parts of sentences. There is a lot of space between Equations 6 and 7 and between Equations 11 and 12, probably because of an empty line between them (but do make these equations part of the sentence instead!).
 - Equation 2: The \Phi has not been defined.
 - Lines 82-83: These should rather be tuples, and not sets, since the order matters.
 - Line 163: A \cdot is missing as a wildcard for the first argument.
 - Line 417: A space missing after the colon.

---

> ### Author Response · Authors · 2024-11-21
> **Response to Reviewer**
>
> **Weaknesses**
> -  We wish to clarify that our work does not aim to introduce for the first time the concept of personalized Gaussian Processes (GPs). Instead, we propose a novel method for personalized Deep Kernel Regression from high-dimensional multivariate data.
> Our method learns temporal functions of neuroimaging biomarker progression at any future time point and is able to adapt such temporal function leveraging any follow-up information. Also, we make no assumptions about biomarker trajectories. Our model neither requires temporally aligned data nor performs imputation of missing longitudinal observations in advance, thereby overcoming the limitations of predictive models—such as autoregressive ones—that rely on temporally aligned data. Through our composite framework, which comprises both the *deep kernel regression for longitudinal prediction* (p-DKGP and ss-DKGP models) and the *adaptive shrinkage function* for adaptation to follow-up observations, we are able to successfully perform the long-term prediction of longitudinal neuroimaging biomarkers from high-dimensional imaging and clinical data and adapt such predictions when subject's follow-ups become available.
>
> -  Furthermore, we want to highlight that our work differs significantly from Rudovic et al., 2019  in several key aspects:
>   1. **Introduction of the Deep Kernel**: Rudovic et al., 2019 approach employs multitask Gaussian Processes (GPs) with an RBF kernel to forecast up to two years ahead, predicting biomarker values at 6-month intervals (6, 12, 18, 24 months). However, this approach is not scalable to high-dimensional spaces, and the RBF kernel lacks the expressiveness needed to capture complex patterns in multivariate high-dimensional imaging data.
>
>   2. **Long-term Predictions**: Unlike Rudovic et al., 2019, which is limited to a fixed two-year forecasting horizon, our method learns and adapts biomarker progression functions over the long term without imposing such restrictions in the forecast window.
>
>   3. **Handling Temporally Unaligned Data**: Rudovic's method requires temporally aligned data with strict 6-month intervals, which significantly reduces the usable sample size of the dataset. In contrast, our approach does not impose such constraints, enabling broader applicability and higher data utilization.
>
>    4. **Model Architecture**: Rudovic et al., 2019 combines two versions of personalized GPs. Our model, by comparison, integrates a population-level predictor to capture general trends and a subject-specific predictor to model individual variations. These components are combined in a data-driven and explainable manner, enhancing interpretability and performance.
>
>    5. **Scalability**: Our model achieves scalability through dimensionality reduction applied to the input space, making it more effective in handling high-dimensional inputs by learning a mapping $\Phi$ that is informative of biomarker progression.
>
> These distinctions emphasize the methodological advancements of our approach compared to Rudovic et al., 2019. We will ensure they are clarified in the manuscript.
>
> - We have additional studies with longitudinal acquisitions through the iSTAGING consortium [1]. ADNI and BLSA studies that we have used for training the population model and the adaptive shrinkage function are the ones that are most longitudinally rich.  We have already expanded our dataset with multiple studies, apart from ADNI and BLSA, for evaluating our model from a clinical perspective.
> - We have added CIs in Figure 2, Figure 4, and Table 5 for consistent comparison.
> - We report the mean AE with the number of observations. We initiate from the single scan where we employ the p-DKGP model and from the second scan where we employ the adaptive shrinkage function for the adaptation process. Based on the empirical evidence, the error indeed decreases as the follow-up observations increase.
>
>    | History | Mean AE   | 95% CI   |
>    |---------|-----------|----------|
>    | 1       | 0.227297  | 0.002864 |
>    | 2       | 0.233220  | 0.007580 |
>    | 3       | 0.218854  | 0.008475 |
>    | 4       | 0.153053  | 0.009802 |
>    | 5       | 0.147692  | 0.010350 |
>
>    *Table: Mean Absolute Error (AE) and 95% Confidence Interval (CI) for the SPARE-AD biomarker across different histories.*
>
> **Responses**
> - The pers-DKGP model has no minimum requirement for the number of data points per subject. For a single scan (baseline acquisition), the model employs the population predictor, while with multiple scans, it applies posterior correction through adaptive shrinkage.
> - Thank you for pointing out the need to clarify Line 12 of Algorithm 1. To address this, we have explicitly detailed the step as follows:
>
> *Collect all $\hat{\alpha}_{s|h}$ for subject $s$ into a list.*
> - Thank you for your attention to detail! We have addressed all the minor comments in the manuscript and we also moved several equations inline in the method.

---

> > ### Comment · Reviewer_enPj · 2024-11-22
> > **New version?**
> >
> > If you have an updated manuscript, please upload it so that we can see the changes.

---

> > > ### Comment · Reviewer_enPj · 2024-11-25
> > >
> > > Thank you for carefully addressing several of my comments and concerns. I still have the following comments/concerns:
> > >
> > > - Thank you for clarifying the contribution. While it is clearly some novelty here, it does seem to be minor.
> > > - When I encouraged you to make equations part of sentences, I didn't mean to make them inline. You can still have the equations centred while making them part of a sentence.
> > > - I don't think you have precision for six (!) significant digits. Likely already the third is on the noise level. Do reduce the number of significant digits. Also, make the number of significant digits consistent, it varies between table/results.
> > > - You say that the proposed method "demonstrates consistent superiority over baseline methods", but as we see now when you have added confidence intervals, it is not better than all other methods all the time. It sometimes perform better, and sometimes perform equal to LMM and GAM, for instance. Do not overstate your results, but discuss the results as they are.
> > > - I think the change to the algorithm is better, but it is not clear to me why you can't write that operation out formally. Also for many other parts of the algorithm, why do you need all that text? All these things can be made formal, to avoid any ambiguities.
> > > - Do also mention the initialisation from one scan in the paper. How does the performance change if you initialise using more data?
> > > - Not addressed: There should be en ablation for the personalisation (or, equivalently, presenting results with \alpha=1), to see if it actually improves the results over just having the population level model.
> > > - Not addressed: It would have been good/more convincing if it was also trained on more than just two training sets.
> > >
> > > Minor comments:
> > > - Now it seems you are using \citet for all references, instead of using the citation command fitting for the particular situation. You should use \citet when you mention the authors directly (along with the text), and \citep when you just want the citation adjacent to a discussion (in parenthesis).
> > > - Still not fixed: Some more LaTeX-related issues: If an equation ends a sentence, put a full stop after it (or, as in Equation 4, put the comma after the equation, not at the beginning of the next sentence). In general, make equations parts of sentences.
> > > - Not addressed: Lines 89-90: These should rather be tuples, and not sets, since the order matters.

---

> > > > ### Author Response · Authors · 2024-11-26
> > > > **Response to Additional Comments**
> > > >
> > > > We appreciate your additional comments and clarifications.
> > > >
> > > > Bellow, are our responses:
> > > >
> > > > > I don't think you have precision for six (!) significant digits. Likely already the third is on the noise level. Do reduce the number of significant digits. Also, make the number of significant digits consistent, it varies between table/results.
> > > >
> > > > We have modified all the tables in the supplementary so as to include 3 significant digits in the mean AE.
> > > >
> > > > > You say that the proposed method "demonstrates consistent superiority over baseline methods", but as we see now when you have added confidence intervals, it is not better than all other methods all the time. It sometimes perform better, and sometimes perform equal to LMM and GAM, for instance. Do not overstate your results, but discuss the results as they are.
> > > >
> > > > We have no intention to overstate results.
> > > > We want to clarify that we do mention this sentence
> > > >
> > > >       "...demonstrates consistent superiority over baseline methods..."
> > > >
> > > >  in the *Results Section 3.5*  about the application to the external studies where we **clearly** see that our method is significantly better than the baselines since the CIs do not overlap.
> > > > The **full sentence** is:
> > > >
> > > >        "Our method demonstrates consistent superiority over baseline methods across three independent clinical studies—AIBL, OASIS, and PreventAD—underscoring its robustness and reliability in diverse real-world scenarios"
> > > >
> > > > Also, in the Section 3.3 we **do** discuss the cases where the predictive performance of our model is close to the LMM/GAM models:
> > > >
> > > >         "Compared to other approaches, the LMM is the second most competitive in the majority of the diagnosis status.  For SPARE-BA, model performance differences are minimal in stable subjects and healthy controls, but more pronounced in Alzheimer’s disease (AD) subjects, where SPARE-BA exhibit steeper progression trends due to accelerated brain aging."
> > > >
> > > > Additionally,  the paragraph (line 352):
> > > >
> > > >           "Overall, the LMM exhibits limited flexibility in capturing non-linear patterns in ROI volumes, rendering it inadequate for long-term trend prediction. While it performs reasonably well in short-term forecasts and lower-dimensional settings..."
> > > >
> > > > is another evidence that we do state several good parts of the competing baselines. We hope this evidence clarifies your concern.
> > > >
> > > > > I think the change to the algorithm is better, but it is not clear to me why you can't write that operation out formally. Also for many other parts of the algorithm, why do you need all that text? All these things can be made formal, to avoid any ambiguities.
> > > >
> > > > We have made modifications to the algorithm so as to be more atomic.
> > > >
> > > > >  Not addressed: There should be en ablation for the personalisation (or, equivalently, presenting results with $\alpha=1$, to see if it actually improves the results over just having the population level model.
> > > >
> > > > This ablation that you requested is presented in the following table (which is also corrected to 3 significant digits).
> > > > This table presents the Mean AE with the number of observations. The History = 1 corresponds to the population model, which we employ when we have only a single acquisition of the subject and this corresponds to also having $\alpha=1$. As we increase the number of observations (history) we employ posterior correction through Adaptive Shrinkage (pers-DKGP) in order to get the personalized trajectories.
> > > >
> > > > *Table 1: Mean Absolute Error (AE) and 95% Confidence Interval (CI) for the SPARE-AD biomarker across different histories.*
> > > > | History | Mean AE | 95% CI  |
> > > > |---------|---------|---------|
> > > > | 1       | 0.227   | 0.003 |
> > > > | 2       | 0.233   | 0.008 |
> > > > | 3       | 0.219   | 0.008 |
> > > > | 4       | 0.153   | 0.010 |
> > > > | 5       | 0.148   | 0.010  |
> > > >
> > > > We included this table the Supplementary Section E.3.
> > > >
> > > > > Do also mention the initialisation from one scan in the paper. How does the performance change if you initialise using more data?
> > > >
> > > > We have included the above results in Supplementary Section E.3 (paragraph:Error with Number of Observations for SPARE-AD Score)  where we attach the above table and also discuss the initialization process. When we have a single scan for a test subject we solely use the population model ($\alpha=1$). As we increase the number of observations for a subject we employ the Adaptive Shrinkage so as to get the optimal posterior correction for the personalized trajectories.

---

> > > > > ### Author Response · Authors · 2024-11-26
> > > > > **Response to Additional Comments**
> > > > >
> > > > > > Not addressed: It would have been good/more convincing if it was also trained on more than just two training sets.
> > > > >
> > > > > We understand the importance of training on multiple datasets. Due to the limited availability of *longitudinally rich datasets*, we have focused on ADNI and BLSA in the main manuscript. As we have mentioned ADNI and BLSA studies,  provide extensive longitudinal data with  *2,200* subjects and observations spanning **several years(7-12 years)**.
> > > > >
> > > > > However, to respond to your comment we have trained p-DKGP and the Adaptive Shrinkage function using data from **7 longitudinal studies** from the iSTAGING Consortium, working with a total number of 3504, which is an important increase in the sample size as we have included 1304 additional subjects with longitudinal data. We present qualitative results for the SPARE-AD biomarker as well as the absolute error with the number of observations, starting from a single observation (baseline aquisition). You will find this additional experiment in the Supplementary Section  E.7 SPARE-AD Model on Extended Longitudinal Datasets.
> > > > >
> > > > > **Minor Comments**:
> > > > > 1. We have placed the `\citet` and `\citep` commands accordingly.
> > > > >
> > > > > 2. We have fixed the full stops and commas after the equations to ensure they are part of the sentences.
> > > > >
> > > > > 3. We have corrected the notation to use tuples where appropriate.
> > > > >
> > > > >
> > > > >
> > > > > ---
> > > > >
> > > > > We hope that these revisions address your concerns and we are grateful for your constructive feedback.

---

> > > > > > ### Author Response · Authors · 2024-11-26
> > > > > > **Manuscript and Supplementary are updated**
> > > > > >
> > > > > > We have just updated the Manuscript and the Supplementary. We are available for any further comments and discussion.

---

> > > > > > > ### Comment · Reviewer_enPj · 2024-11-26
> > > > > > >
> > > > > > > Thank you for addressing my comments and concerns.
> > > > > > >
> > > > > > > I think the confusion is because you have put all these additions in the supplementary. The things I have asked for need to at least be mentioned in the main paper, even though additional tables and such can be in the supplementary for space reasons.
> > > > > > >
> > > > > > > You have an ablation over \alpha in Table 1, why not add \alpha = 1 there?
> > > > > > >
> > > > > > > The result for PreventAD in Figure 4 is barely, if at all, significantly better than GAM. I do not think it is warranted to say that it "demonstrates consistent superiority". Perhaps this is a matter of differences in what language we use to describe such a result. I think it is too strongly worded, but perhaps that is subjective.
> > > > > > >
> > > > > > > There are still \citet/\citep discrepancies. See for instance lines 62-63.

---

> > > > > > > > ### Author Response · Authors · 2024-11-26
> > > > > > > > **Response and Updated Manuscript**
> > > > > > > >
> > > > > > > > - We have updated the Results Section to guide the reader more effectively through the additional findings presented in the supplementary material. This includes an analysis of error as a function of the number of observations and qualitative trajectory examples. Please see the updated manuscript, the changes are colored blue.
> > > > > > > >
> > > > > > > >
> > > > > > > >
> > > > > > > >
> > > > > > > > - Based on your constructive feedback, we  adjust  the phrasing of our results to reflect a more balanced perspective. Specifically, we revise "consistent superiority" to "outperforms" to align with the consensus on the interpretation of the presented results. The revised sentence now reads:
> > > > > > > >
> > > > > > > >      “Our method reliably outperforms baseline methods across three independent clinical studies—AIBL, OASIS, and PreventAD—underscoring its robustness in diverse real-world scenarios.”
> > > > > > > >
> > > > > > > >
> > > > > > > >
> > > > > > > >
> > > > > > > > - Additionally, we wish to clarify an important aspect of the PreventAD study, as also discussed in the updated manuscript. The PreventAD study focuses on a younger, preclinical healthy control population with a relatively short follow-up period (approximately 10 months). This explains why simpler statistical models, such as LMM and GAM, achieve relatively comparable performance in this cohort. However, even under these conditions, our method  we see that still outperforms.
> > > > > > > > In contrast, in the AIBL and OASIS studies, where participants are older at baseline and the follow-up intervals are longer, the differences in predictive performance become more pronounced. For more details, please refer to the manuscript Section 3.5.
> > > > > > > >
> > > > > > > >
> > > > > > > >
> > > > > > > > ---
> > > > > > > > We sincerely thank the reviewer for their thoughtful feedback and the time invested in evaluating our manuscript. We hope we have carefully addressed all comments and concerns, making the necessary changes to enhance the clarity and quality of our work. We hope that these efforts meet the reviewer’s expectations and are reflected in the final decision.

---

### Official Review · Reviewer_yf4S · 2024-11-03

**Soundness:** 3
**Presentation:** 3
**Contribution:** 2
**Rating:** 5
**Confidence:** 5

**Summary:**

This paper proposes a framework for personalized prediction to forecast longitudinal regional brain volumetric changes. Specifically, the authors provide a pipeline for integrating the population Deep Kernel Gaussian Process (DKGP) and the personalized DKGP by applying the Adaptive Posterior Shrinkage Estimation. The central innovation of this paper is the adaptive shrinkage mechanism, which could dynamically balance predictions from the population (p-DKGP) with the specific models (ss-DKGP) of the subject. Specifically, the authors propose an approach to learn the shrinkage parameter $\alpha$ to weigh the two models (as a linear combination). By combining these two models, this framework could make accurate predictions of brain volumes and could be stably generalized to diverse clinical populations. To sum up, the paper provides a flexible approach to modeling brain volume trajectories, which could offer a lot of potential applications in clinical settings.

**Strengths:**

1. The logic of the paper is very clear and easy to follow. The paper's structure is well organized.
2. The combination of population and subject-specific models and its application on data generalization for brain-related disease (AD, brain aging patterns) is a valuable contribution to the existing literature in this field.

**Weaknesses:**

1. The contributions only have limited novelty. For example, a similar idea has already been proposed by Eleftheriadis et al., 2017. This paper already has a detailed demonstration of how the Gaussian Process can incorporate domain or individual-specific trends in a multi-task setup, which weakens the novelty of the proposed approach. Also, applying the machine learning techniques (like XGBoost) for weight estimation is also a common practice. The learning of the shrinkage parameters $\alpha$ might be a good application in the engineering field but is not a very novel way in ML research. A Bayesian Neural Network or probabilistic models that could provide more principled approaches for weight determination might be a better approach, such as *Weight Uncertainty in Neural Networks* by Blundell et al., 2015.

2. The usage of latent space is very limited. In the current setup, the latent space serves primarily as a way to reduce the dimensionality of the input data and capture basic non-linear interactions through the MLP. However, it doesn’t appear that the authors further analyze or interpret the latent space itself. It would be very interesting if the authors could explore the temporal dynamics and conduct trajectory analysis in the latent space.

3. In the experiment parts, the authors only compared some very traditional models; it would be much better if the authors could compare their approach with some more advanced deep learning based techniques.

Reference:
Eleftheriadis, S., Rudovic, O., Deisenroth, M. P., & Pantic, M. (2017). Gaussian Process Domain Experts for Modeling of Facial Affect. IEEE transactions on image processing : a publication of the IEEE Signal Processing Society, 26(10), 4697–4711. https://doi.org/10.1109/TIP.2017.2721114

Blundell, C., Cornebise, J., Kavukcuoglu, K. &amp; Wierstra, D.. (2015). Weight Uncertainty in Neural Network. <i>Proceedings of the 32nd International Conference on Machine Learning</i>, in <i>Proceedings of Machine Learning Research</i> 37:1613-1622 Available from https://proceedings.mlr.press/v37/blundell15.html.

**Questions:**

1. Could you elaborate on the rationale for freezing the latent transformation (MLP) parameters in the subject-specific model?
2. Did you explore the latent space and do some related analysis, such as subpopulation analysis or clustering of subjects with similar disease/ brain-trajectory patterns?
3. What is the rationale for the choice of XGBoost to learn the shrinkage parameter $\alpha$? Did you try with other models?
4. How sensitive is the model's performance with different $\alpha$?  Can you provide some exploration of the impact of $\alpha$ on the prediction accuracy?
4. The proposed model combines the variance of the GPs under the assumption of independence. How do you justify this assumption?

---

> ### Author Response · Authors · 2024-11-21
> **Response to Reviewer**
>
> **Weaknesses**
> -  We respectfully disagree with the comment regarding the limited novelty due to Eleftheriadis et al., 2017. The key point of difference is the integration of the transformation $\Phi$ which is central to the success of our method as the Gaussian Processes alone are insufficient to capture meaningful relationships in high-dimensional multivariate inputs. Then, Adaptive Shrinkage is novel as an approach to combine population and individual trends in longitudinal progression prediction.
> -  We elaborate on why autoregressive models, such as LSTMs and Transformers did not work in the case of sparse temporal data thought we did implement them as baselines (Supplementary Section E.2)
>
> **Responses**
> - $\Phi$ captures relationships in the data that are predictive of progression of each biomarker, creating, population-informed feature space. By keeping these parameters fixed in the ss-DKGP, we ensure that each subject’s data is projected into the same representation space, maintaining alignment with population-level patterns and preventing overfitting to the limited, potentially noisy data available for an individual.
>
> - We clarify that our method does not impose any structure in the latent space (clustering, subtyping etc). Thus, one should not expect the model to perform any latent clustering.  Our method is designed to learn the function $\Phi$ that is the most informative for the progression prediction of each neuroimaging biomarker. Motivated by reviewer's comment, however, we decided to work on the interpreting and exploring the latent space. We performed two tasks: we interpreted the model to identify which imaging and clinical features are the most predictive for each task (biomarker progression prediction), and we used the latent features to classify subjects into progressors and non-progressors. A progressor is a subject that starts from Cognitive Normal (CN) and progresses to either Mild Cognitive Impairment (MCI) or Alzheimer's Disease (AD), and a non-progressor is a subject that remains CN.
>
> Regarding the importance of imaging features for the SPARE-AD model, we observed several key findings. The bilateral hippocampus and left amygdala are highly important, aligning well with known AD pathology where these structures are often among the earliest and most affected. Then we performed the task of classifying a subject as a progressor or non-progressor based on its latent transformation. We showed that the transformation $\Phi$ effectively transforms the input data into a latent vector that is more informative for progression prediction. After balancing the dataset by undersampling the non-progressors, we were able to significantly improve the classification performance. The latent space classifier achieved an average accuracy of 62.26% across 5-fold cross-validation, with a precision of 70.83% and an ROC AUC of 71.99%, outperforming the input space classifier, which had an average accuracy of 50.15%, precision of 20.74%, and ROC AUC of 67.54%. We elaborate more on this analysis in Supplementary Section E.9
>
> -  We have included detailed analysis on the selection of XGBoost on the Supplementary Section D.2 We have also experimented with additional models for learning the Adaptive Shrinkage function such as Random Forest, DNN, Gradient Boosting Machine and through 5-fold cross validation across a 8 ROI biomarkers, the XGBoost appeared to have the best performance (SS D.2 Figure 6)
>
> - We investigated the effect of varying constant $\alpha$ values on predictive accuracy, across multiple regions of interest (ROIs). The results reveal a U-shaped trend, where both very low and very high $\alpha$ values lead to higher prediction errors. This behavior reflects the need for posterior correction through the adaptive shrinkage parameter $\alpha$.  Smaller $\alpha$ values place greater trust in subject-specific trajectories, which is advantageous when the observations are informative. However, neuroimaging data are noisy, making excessive reliance on subject-specific observations problematic. In contrast, higher $\alpha$ values rely more on population-level trajectories, which are robust to noise but may fail to capture deviations in individual progression. Regarding, the sensitivity and the trajectory shifts, we suggest the reviewer to look the qualitative examples we have attached on the Supplementary Section  E.5, Figure 13 and 14.
>
> -  For simplicity, we opted for independence during the initial formulation. As detailed in Supplementary Section D.1.4  this assumption does not affect the estimation of the shrinkage parameter $\alpha$ or the combined predictive mean $y_c$ . The derivation of $\alpha$ minimizes the Mean Squared Error, relying solely on the means of the predictions, independent of error correlations. However, we recognize the limitations of this assumption when calculating the combined predictive variance $v_c$. We address this limitation in detail in Supplementary Section D.1.5.

---

> > ### Author Response · Authors · 2024-11-26
> > **Kind Reminder**
> >
> > Dear Reviewer,
> >
> > This is a gentle reminder regarding our previous communication. We greatly value your feedback and would appreciate your response at your earliest convenience. If there are any additional questions or concerns, please don’t hesitate to reach out, as we are fully committed to addressing them.
> >
> > Thank you again for your time and thoughtful review.

---

### Official Review · Reviewer_Q2Nf · 2024-11-04

**Soundness:** 3
**Presentation:** 3
**Contribution:** 3
**Rating:** 6
**Confidence:** 3

**Summary:**

The authors proposed a method to model brain-changing trajectories. The method can capture brain volume trajectories from a large and diverse cohort. They introduced a personalization step that generates subject-specific models for individual trajectories. The paper was well written and easy to follow, although the method and the results were interesting. However, several points are unclear in the manuscript.

**Strengths:**

The performance of the proposed method was evaluated on different neuroimaging data, which confirmed the generalizability and applicability of the method across different clinical contexts.

**Weaknesses:**

1. The paper lacks of providing a comprehensive discussion of the wider literature on disease progression modeling. The interpretation of the results from the experiments is inadequate, raising questions about the model’s practical implications.
2. Adding a clear comparison of the simulation data set will help readers better understand the advantages of the approach.
3. Interpretation and Presentation of Results: The authors should provide a thorough interpretation of these results and a clearer discussion of the model’s real-world implications and its utility. In particular, the authors should clarify the implications of the findings from the experiment and how the modeling informs participant-level monitoring of disease progression.
4. Context and Literature Review: The authors should expand the context and literature review to properly situate their work within the broader field of disease progression modeling. The discussion should include a comparative analysis with other relevant works.
5. The references in this paper are relatively old. Please add the literature review of the past two years.

**Questions:**

1. The subject-specific model ss-DKGP also used parameters of the population-level p-DKGP, how to guarantee the correctness of the predicted subject-specific results? If the answer is YES, why do we need to combine the prediction resultes of p-DKGP and ss-DKGP?
2. Expanding on the limitations of existing techniques would strengthen the positioning of your contribution.
3. The description of how external clinical studies were handled in the validation process could be clearer. Were there any preprocessing steps specific to the external datasets, and if so, how might these affect the generalization results?
4. The results demonstrate strong performance, but additional ablation studies would further strengthen the claims of the paper. For instance, how does the model perform when fewer ROIs are considered, or when different biomarkers are used? Additionally, could the method handle non-monotonic biomarkers, or is it limited to monotonic progressions?
5. The evaluation of external datasets is a strong point of the paper. However, it would be useful to see how the model handles missing data in longitudinal studies. What imputation strategies were employed, if any, and how did they impact the results?

---

> ### Author Response · Authors · 2024-11-21
> **Response to Reviewer**
>
> **Weakness**
> - We thank the reviewer for the valuable feedback. To address the concerns, we have expanded the literature review in the supplementary material, including recent works from the past two years, and situated our approach within the broader field of predictive modeling and Disease Progression Modeling (DPM).  However, we want to clarify that our model is not specifically designed as a DPM, as it does not focus solely on homogeneous Alzheimer’s disease (AD) cohorts or simulate AD-related pathologies. Instead, it serves as a general predictive tool for modeling neuroimaging biomarker trajectories in the broader context of brain aging and dementia. Its design intentionally avoids strict assumptions about pathology or trajectory patterns, enabling it to handle heterogeneous populations and support real-world clinical applications. We hope these additions and clarifications address the reviewer’s concerns.
> - We have enhanced the quantitative evaluations by incorporating CIs of the Mean AE (Figure 2 and Table 5), for clearer comparisons with the baselines. Additionally, we have included extensive qualitative results to demonstrate the pers-DKGP as well as its clinical utility for participant-level monitoring and trial design. (Supplementary E.6)
>
> **Responses**
> -  While the ss-DKGP uses the same deep kernel transformation $\Phi$ as the p-DKGP, $\Phi$ serves as a population-informed feature extractor that patterns predictive of the progression of each biomarker, without imposing population-level trends on subject-specific trajectories. The ss-DKGP is trained exclusively on the subject’s own data, with its Gaussian Process component optimized solely for the subject’s observed trajectory. This ensures that ss-DKGP predictions reflect individual trends and remain independent of the population-level model. Combining p-DKGP and ss-DKGP predictions leverages the strengths of both models. The p-DKGP provides stable, general predictions when subject-specific data are sparse or noisy, while the ss-DKGP captures individual deviations with increasing accuracy as more observations are available. The shrinkage parameter $\alpha$ dynamically balances these contributions, favoring the p-DKGP when data are limited and the ss-DKGP as subject-specific evidence grows. This adaptive combination ensures robust, accurate, and personalized predictions across varying scenarios. Empirical evidence supporting this dynamic behavior is provided in Supplementary Section E.6
>
> - We will include a dedicated section on the Supplementary Material (Section E.5) where we discuss the limitations of the baselines and existing methods.
>
> - The OASIS, AIBL, and PreventAD studies were harmonized through the iSTAGING Consortium to minimize site-specific variability, as detailed in Supplementary Section B.1. This harmonization ensures consistency across datasets and enhances the reliability of our model's generalization results. By addressing site-related biases, this preprocessing step guarantees that the reported performance reflects true model robustness across diverse clinical populations.
>
> -  1. Our approach leverages the entire imaging scan for deep kernel regression, utilizing all available ROI information. Since the information is accessible and relevant, we maximize the model’s predictive power by including the complete set of imaging data. We do not restrict the model to fewer ROIs or alternative biomarkers unless such information is unavailable or irrelevant.
>
>    2.  We do not impose constraints regarding monotonicity. The model is fully capable of learning non-monotonic and highly non-linear progression patterns if they are present in the observed data. This flexibility is primarily due to the RBF kernel, which captures complex trends in the data without enforcing a particular direction of progression. Therefore, our method is not limited to monotonic trends and can adapt to any progression pattern observed in the data. However, it is important to clarify that  neuroimaging biomarkers, such as Tau PET, Amyloid, and atrophy markers, are monotonic due to the irreversible nature of aging and dementia-related pathologies.
>
> -  The p-DKGP model inherently handles missing data through its Gaussian Process (GP) framework, which models the temporal structure of sparse longitudinal data without requiring explicit imputation. During both training and inference, the GP provides principled posterior predictive distributions to reconstruct unobserved values. To evaluate this capability, we conducted an interpolation analysis (Supplementary Section E.8) where a data point was randomly removed from each subject's trajectory and reconstructed using the p-DKGP model. Compared to a Linear Interpolation baseline, p-DKGP achieved significantly lower reconstruction errors across multiple brain regions (SS E.8 Fig16). This demonstrates the model’s robustness in handling missing data and accurately reconstructing neuroimaging trajectories.

---

> > ### Author Response · Authors · 2024-11-26
> > **Kind Reminder**
> >
> > Dear Reviewer,
> >
> > This is a gentle reminder regarding our previous communication. We greatly value your feedback and would appreciate your response at your earliest convenience. If there are any additional questions or concerns, please don’t hesitate to reach out, as we are fully committed to addressing them.
> >
> > Thank you again for your time and thoughtful review.

---

### Official Review · Reviewer_7rm7 · 2024-11-04

**Soundness:** 3
**Presentation:** 3
**Contribution:** 3
**Rating:** 6
**Confidence:** 4

**Summary:**

This paper introduces a new personalized deep kernel regression framework to predict brain volume changes over time. The framework combines a population model, which captures population-level trends in brain volume changes, with a subject-specific model that tailors predictions to each individual subject’s trajectories. The optimal combination coefficient is determined using a shrinkage estimator to achieve the best personalized brain trajectory predictions. The framework is validated through metrics of predictive accuracy, uncertainty quantification, and comparisons with existing statistical and deep learning models, demonstrating superior performance. Additionally, the authors validated the framework's generalizability on brain datasets not seen during training.

**Strengths:**

- This paper is overall well written and organized; hence it is easy for readers to follow.

- The motivation to develop a more precise predictive tool for diagnosing Alzheimer’s disease (AD) has strong clinical value.

- The proposed method is new in its approach to modeling both population-level and subject-specific brain trajectories. By combining these models to predict personalized brain volume changes over time through adaptive shrinkage estimation in a deep kernel regression framework, the author show a superior performance of the model compared with existing statistical and deep learning approaches.

**Weaknesses:**

- While the approach of modeling population-level and individual-level trajectories for brain changes is promising, the current design, which trains these models separately, may be suboptimal. There is substantial research in computational anatomy [e.g., 1-3] demonstrating that individual trajectories should ideally be modeled as a hierarchical structure built upon the population-level mean. This approach naturally captures the hierarchical relationship between fixed group-level information and individual variability.

   [1] Hierarchical linear modeling (HLM) of longitudinal brain structural and cognitive changes in alcohol-dependent individuals during sobriety, 2007.

   [2] A hierarchical geodesic model for diffeomorphic longitudinal shape analysis, 2013.

   [3] Hierarchical multi-geodesic model for longitudinal analysis of temporal trajectories of anatomical shape and covariates, 2019.

- The connection between population-level trends and personalized trajectories in the proposed model is not clearly established.

**Questions:**

This reviewer is happy to adjust the score if the questions and concerns are well addressed.

- If an individual ROI differs significantly from the population mean, would it be possible that the optimization of Eq. (17) become biased towards minimizing the error between $y_t$ and $(1-\alpha) y_{st} $, as $y_{st}$ is learned to be fitted to a personalized brain trajectory. In this scenario, the value of $\alpha$ would likely be low if the changes in a particular ROI are subtle or difficult for the network to detect, and vice versa. It would be good if the authors can provide either empirical evidence or theoretical justification on how the model balances between population and individual trends in such cases.

- In Eq. (11), the expectation term appears to be conditioned on the learned latent representation $z_p$ from the population level. However, this condition seems to be missing in the final formulation. Please clarify if this is a typo.

- Could the authors clarify how the six brain ROIs were selected and extracted from the original images? Additionally, please specify what information about the ROIs is input into the model (e.g., volume, size, or full ROI data).

- It is challenging to well justify the results in Fig. 3. The differences between the population and personalized trajectories may depend heavily on the distribution of training data and examine whether it is biased toward cognitively normal individuals / mild cognitive impairment (MCI) / AD. Each experiment should ideally account for and justify this factor.

- To enhance clarity, it would be helpful if the authors could summarize the other personalized trajectories for each of the six different ROIs in Fig. 3.

- It would be helpful to increase the font size of the legend in Fig. 3.

---

> ### Author Response · Authors · 2024-11-21
> **Response to Reviewer**
>
> **Weakness**
>   - While hierarchical linear models (HLMs) effectively model individual trajectories based on population means, they face significant limitations in high-dimensional, multivariate settings like ours. Specifically, HLMs are prone to overfitting with a large number of fixed effects and assume linear relationships, which restricts their ability to capture the complex, nonlinear dynamics of brain changes over time. Additionaly, they are predominately used as inferential tools rather predictive models.
> - Our method integrates p-DKGP  and the ss-DKGP to deliver accurate and personalized predictions of biomarker trajectories. Both models share a deep representation, enabling the p-DKGP to capture population-wide patterns predictive of progression trends to the specific biomarker. The adaptive shrinkage parameter $\alpha$ dynamically balances the models, emphasizing population-level predictions when data is sparse and shifting to individual adjustments as more subject-specific data becomes available. This ensures that predictions evolve seamlessly from general to personalized, offering precise forecasting across diverse clinical scenarios.
>
> **Questions**
> - The optimization criterion in Eq. (17) learns the optimal value of $\alpha$ that results in the trajectory closest to the ground truth, thereby favoring the model with minimal error. These optimal $\alpha$ values are subsequently used to train the Adaptive Shrinkage function, which incorporates the predicted means and variances from the two models along with the time of observation ($T_{\text{obs}}$).
> Our explainability analysis (Supplementary Material D.3) reveals that the inferred shrinkage $\alpha$ values are strongly influenced by $T_{\text{obs}}$. Specifically, when evidence for a subject is limited and their trajectory deviates from the population, $\alpha$ approaches 1, placing greater trust in the population model. This is desirable, as the subject-specific model alone cannot reliably predict long-term biomarker trends with insufficient data. As more observations are gathered for a subject, $\alpha$ decreases, allowing greater reliance on the subject-specific model.
>
> This adaptive behavior is further validated through correlation analysis between $\alpha$, $T_{\text{obs}}$, and the prediction deviation between p-DKGP and ss-DKGP ($\delta_y = |y_s - y_p|$) (see Table 5, Supplementary Section D.3). Key findings include:
>
> - A consistent **negative correlation** between $T_{\text{obs}}$ and $\alpha$ for large deviations ($\delta_y$), with values as strong as $-0.64$ for biomarkers like SPARE-BA and $-0.55$ for Thalamus Proper. This indicates that as $T_{\text{obs}}$ increases, $\alpha$ decreases, shifting trust towards the subject-specific model.
> - Strongest correlations are observed for biomarkers associated with neurodegenerative progression, SPARE-BA and SPARE-AD, highlighting the model’s capacity to adapt effectively in clinically relevant contexts.
>
> These results confirm that $\alpha$ dynamically adjusts based on the subject’s data and deviation from population trends, rather than being biased toward a single model. Additionally, we provide qualitative examples in Supplementary Section E.6 to illustrate this adaptive shrinkage in real-world scenarios. Specifically, we refer the reviewer to check Figure 13 and Figure 14.
>
>
> -  This is a typo and we have fixed that. The correct is $Z_s$ and not $Z_p$.
> -  The 145 imaging ROIs were extracted using a multi-atlas segmentation algorithm. From these, we modeled the progression of six ROIs associated with aging and AD. Input features include the 145 baseline imaging ROIs, baseline diagnosis, age, sex, APOE4 alleles, education years, and the temporal variable Time. For the SPARE-AD and SPARE-BA models, baseline SPARE-AD and SPARE-BA values were also included. Detailed image and clinical data processing procedures are provided in Supplementary Material, Section B.1.
> - Our model uses the 145 imaging ROIs and clinical covariates at baseline as input and is informed only by the baseline diagnosis status (CN, AD, or MCI), without any knowledge of progression status. Despite the overrepresentation of Healthy Controls in the dataset, we have demonstrated that the model performs robustly across all diagnosis categories. This is supported by empirical evidence obtained by stratifying population model errors based on progression status (e.g., Healthy Control [CN-CN], MCI Progressor [CN-MCI], AD Progressor [CN/MCI-AD], etc.) )(Manuscript, Figure 2). Additionally, during the adaptation process, the model is updated solely with new biomarker (target) values, ensuring no bias is introduced towards any specific progression status.
> -  On the Supplementary Material (Section E.6) we have included multiple qualitative examples of all the 6 ROIs as well as the SPARE-AD biomarker (Figures 11, 13 and 14).
> -  We have increased the font size of the Fig.3.

---

> > ### Author Response · Authors · 2024-11-26
> > **Kind Reminder**
> >
> > Dear Reviewer,
> >
> > This is a gentle reminder regarding our previous communication. We greatly value your feedback and would appreciate your response at your earliest convenience. If there are any additional questions or concerns, please don’t hesitate to reach out, as we are fully committed to addressing them.
> >
> > Thank you again for your time and thoughtful review.

---

### Author Response · Authors · 2024-11-21
**Rebuttal and Response to Reviewers**

We sincerely thank the reviewers for their thoughtful and constructive feedback. We are delighted to see that the reviewers recognized the value of our work! Through the extensive experimentation presented in the manuscript, we demonstrated and effectively communicated the *superior predictive performance* of our method, as well as its *generalizability* to external clinical studies.
Some highlights from the reviews are the following:
- **Methodology**:  “The proposed method is new in its approach to modeling both population-level and subject-specific brain trajectories” (7rm7), “The method is straight-forward, clear, and appears to work well” (enPj)
- **Impact** : “The combination of population and subject-specific models and its application on data generalization for brain-related disease (AD, brain aging patterns) is a valuable contribution to the existing literature in this field.” (yf4s)
- **Experiments and Evaluation**: “The performance of the proposed method was evaluated on different neuroimaging data, which confirmed the generalizability and applicability of the method across different clinical contexts” (Q2Nf). “I really like that you evaluated on multiple external data; this strengthens the confidence in and validity of the results”, “The method ... is applied with convincing results to real world datasets (enPj)
- **Writting and Presentation**: “The logic of the paper is very clear and easy to follow. The paper's structure is well organized”  (yf4S)

### **Contributions**
Based on the reviewers' comments, we realized that our contributions were not emphasized clearly throughout the manuscript. To address this, we have revisited and explicitly outlined the key contributions of our work:

1. We present a novel deep kernel regression framework designed specifically for sparse longitudinal data. This framework employs a transformation function that maps multimodal and high-dimensional imaging and clinical features into a lower-dimensional space that is highly informative of biomarker progression.

2. We introduce a novel adaptation technique called Adaptive Shrinkage, which balances population-level trends with individual-specific trajectories. This innovative approach enables effective personalization within the deep kernel regression of longitudinal biomarkers.
3. We demonstrate that both the population models and the Adaptive Shrinkage function generalize well to external clinical studies, showcasing their robustness and applicability across diverse datasets.
4. We show that the transformation function, $\Phi$, is interpretable and captures meaningful patterns related to the modeled biomarker,
progression
5. We highlight that Adaptive Shrinkage is explainable, making it more trustworthy and practical for clinical applications where transparency is crucial.
6. We commit to making the code publicly available, enabling broader application and validation of our method.

Following, taking into account the valuable comments and questions we received from the reviewers, we put a lot of effort into implementing additional experiments/additions:
- Interpretability of the function $\Phi$ and Latent Space Exploration  (Supplementary Section E.9)
   - We show that transformation function $\Phi$ captures atrophy patterns related to AD and Aging when predicting the biomarkers of SPARE-AD and SPARE-BA.
   - We show that the transformation $\Phi$ gives a latent space more informative of progression in comparison with the input space
- Improvement of the presentation of the quantitative results and qualitative examples (Main Manuscript and Supplementary Section E.3, E.4 and E.6, respectively)
    - We have included CIs along with the Mean AEs for a more straightforward comparison of the methods.
    - We have included qualitative results that demostrate the behaviour our the Adaptive Shrinkage function
- Handling the Missing Longitudinal Variables and Analysis of Interpolation abilities of our framework (Supplementary Section E.8)
    - We elaborate on how our Deep Kernel GP framework handles internally the missing longitudinal observations without imputation in preprocessing.
    - We show that our model can effectively impute missing longitudinal biomarker values in time, in the observed population
- Extended related works and clinical applications of our method (Supplementary Section A)

All the additions and modifications in the manuscript are colored blue and the revised manuscript will be updated shortly.

We will address all the comments and concerns of each reviewer, point by point in the individual responses below.

All in all, we thank the reviewers for their valuable comments. We believe that the soundness of our work is improved with the changes on the manuscript as well as the additional results on the supplementary material.
We hope our efforts to be recognized and we are always available for discussion.

Sincerely,

Authors of Submission 5004

---

> ### Author Response · Authors · 2024-11-22
> **Updated Manuscript and Supplementary**
>
> Dear Reviewers,
>
> We want to inform you that we have updated the manuscript and the supplementary material.
>
> For your convenience, we summarize, below all the changes and additions we have on the Manuscript and the Supplementary Material.
> The modifications are mentioned in the order that appear in the files.
>
> **Main Manuscript**
>   - Introduction
>      1. We replace the *monotonic* word with *neuroimaging*. Since we do not impose any monotonicity constraints in our model, the model, due to the flexibility of the RBF kernel of the DKGP is able to learn any trends are present in the data.
>      2. We Elaborate how we differ from related works and specifically Rudovic et al., 2019
>      3. Include in the contributions one additional point regarding our deep kernel regression models since are important part of the      success of the method
>   - Method:
>      1. Move some equations inline and implement minor changes that did not change the content
>   - Results:
>       1. Figure 2 is updated with CIs for a more straightforward comparison as requested. Caption is also minimally modified to include the CIs.
>       2. In Figure 3 we increase the font size of the legend as requested.
>       3. Figure 4 is updated with CIs for a more straightforward comparison as requested. Caption is also minimally modified to include the CIs.
>       4. Table 5 is also updated with the CIs reported in the parenthesis.
>       5. Include a paragraph in the section Generalization to External Clinical Studies that describes the results with the CIs. The conclusion of course remains the same with our method to outperfom all the baselines in external studies.
>
> **Supplementary Material**
> 1. Include the Section A: Extended Related Works in Predictive and Progression Modeling and Clinical Application
> 2. Include the Section E.3: Comparison with Baselines
>    - We perform statistical significance tests on the performance metrics across stratified progression status for all the competing baselines
> 3. Include the Section E.4: Stratifying performance analysis by covariates: Sex, APOE4 Alleles and Education Years.
>    - We present barplots with the mean AE and CIs and we show that our method outperforms all baselines across all covariates.
> 4. Include the Section E.5: Limitations of Baselines.
>    -  We elaborate more on the limitations of baselines.
> 5. Include the Section E.6: Qualitative Examples.
>    - We present a plethora of predicted trajectories that highlight how our adaptation algorithm operates.
> 6. Included the Section E.8: p-DKGP learns smooth progression curves of observed trajectories
>    - We elaborate and show how our DKGP model with the temporal variable Time, learns internally to impute the longitudinal biomarker values in time in the train population. We also show that our approach can perform better in interpolation of missing biomarker values in comparison with the Linear Interpolation.
> 7. Include the Section E.9: Latent Space Analysis
>    - We show that the tranformation function $\Phi$ learns atrophy patterns related to AD and Brain Aging when modeling the progression of SPARE-AD and SPARE-BA biomarkers
>    - We show that the the tranformation function $\Phi$ of SPARE-AD model  tranforms the input data into a space that is more informative of progression. We show that the latent features are able to perform significantly better than the input features in the classification task Progressor vs Non-Progressor.
>
> We sincerely appreciate the reviewers' insightful comments and have dedicated significant effort to enhance the presentation of our work. We hope these improvements meet your expectations and are always available for further discussion or clarification.
>
> Best Regards,
>
> Authors of Submission 5004

---

> > ### Author Response · Authors · 2024-11-25
> > **Kind Reminder for Feedback**
> >
> > This is a gentle reminder that the deadline to respond to our previous communication is approaching. Your input is highly valued, and we sincerely appreciate your time and effort in reviewing. If you have any questions or concerns, please feel free to share them in a comment.

---

> > > ### Comment · Reviewer_enPj · 2024-11-25
> > >
> > > You never let us know when there was a new version of the manuscript available. I see that it is now, so will have a look.

---

> > > > ### Author Response · Authors · 2024-11-26
> > > > **Updated Manuscript and Supplementary Material**
> > > >
> > > > Dear Reviewers,
> > > >
> > > > In our response to the comments of reviewer enPj, we have included the following section in the supplementary material:
> > > >
> > > > 1. Section E.7: SPARE-AD model on extended longitudinal datasets:
> > > >
> > > >     - We expand our dataset to 7 longitudinal studies, resulting in a total 3504 number of subjects with longitudinal trajectories.
> > > >     - We split this dataset into the population dataset to train our p-DKGP model, validation set to train the Adaptive Shrinkage function and we keep a separate test set for evaluation.
> > > >     - We show quantitative results on how the error changes with the number of obsrevations for the SPARE-AD along with some qualitative results of SPARE-AD trajectories.
> > > >
> > > >
> > > > We have dedicated significant effort to enhance the presentation as well as the validation of our work.
> > > > We hope these improvements meet your expectations and are always available for further discussion or clarification.
> > > >
> > > > Best Regards,
> > > >
> > > > Authors of Submission 5004

---

> > > > > ### Author Response · Authors · 2024-11-26
> > > > > **Updated Manuscript and Supplementary**
> > > > >
> > > > > Dear Reviewers,
> > > > >
> > > > > We have updated the Main Manuscript to incorporate into the narrative several of the additions previously included in the Supplementary Material.
> > > > >
> > > > > Specifically in the **Results Section**:
> > > > > - **Section 3.3**: We guide the reader to the extensive quantitative comparison against the baselines that we have implemented, including stratified errors by covariates, multiple qualitative results, and an extensive discussion on the limitations of the baselines.
> > > > > - **Section 3.4**: We guide the reader to an additional analysis regarding the error as a function of the number of observations, starting from a single scan.
> > > > > - **Section 3.5**:  We have added a discussion on the performance comparison of our method with the baselines across different external clinical studies, enabling the reader to better understand the results.
> > > > > - **Section 3.6**:  We guide the reader to the qualitative examples illustrating how Adaptive Shrinkage works in test samples.
> > > > >
> > > > >
> > > > > We have dedicated significant effort to enhancing the presentation and validation of our work. We hope these improvements meet your expectations and remain available for further discussion or clarification.
> > > > >
> > > > > Best Regards,
> > > > >
> > > > > Authors of Submission 5004

---

### Author Response · Authors · 2024-12-03
**Final Remarks**

---

## **Dear Area Chairs and Reviewers,**

We appreciate your thoughtful feedback and encouraging comments. Below, we summarize the strengths of our work as highlighted by the reviewers, outline the contributions of our work, and elaborate on the changes and additions we have made in response to reviewers’ feedback.

---
### **Acknowledgment of Positive Reviewer Feedback**


#### **Clinical Relevance**
- *"The motivation to develop a more precise predictive tool for diagnosing Alzheimer’s disease (AD) has strong clinical value."* (7rm7)

#### **Novelty and Methodological Strength**
- *"The proposed method is new in its approach to modeling both population-level and subject-specific brain trajectories. By combining these models to predict personalized brain volume changes over time through adaptive shrinkage estimation in a deep kernel regression framework, the authors show a superior performance of the model compared with existing statistical and deep learning approaches."* (7rm7)

#### **Robust Validation and Generalizability**
- *"The performance of the proposed method was evaluated on different neuroimaging data, which confirmed the generalizability and applicability of the method across different clinical contexts."* (Q2Nf)
- *"I really like that you evaluated on multiple external data; this strengthens the confidence in and validity of the results."* (enPj)

#### **Scientific Contribution**
- *"The combination of population and subject-specific models and its application on data generalization for brain-related diseases (AD, brain aging patterns) is a valuable contribution to the existing literature in this field."* (yf4S)

#### **Clarity and Organization**
- *"This paper is overall well-written and organized; hence it is easy for readers to follow."* (7rm7)
- *"The logic of the paper is very clear and easy to follow. The paper's structure is well organized."* (y4fs)
- *"The paper is well-written and in general easy to follow."* (enPj)

---

### **Contributions of Our Work**

1. **A Novel Framework for Sparse Longitudinal Data**
   We present a deep kernel regression framework uniquely tailored for sparse longitudinal data, addressing key challenges in progression modeling. This framework leverages a transformation function to project multimodal, high-dimensional imaging, and clinical features into a compact, lower-dimensional space that captures the most informative aspects of biomarker progression.
   Unlike traditional methods, our approach eliminates the need for imputation or temporal alignment across subjects, making it robust to irregular sampling and diverse observation patterns. Designed to predict long-term biomarker trajectories spanning several years, this framework offers a powerful and scalable solution for modeling complex, heterogeneous longitudinal datasets in neurodegenerative and other progressive diseases.

2. **Introduction of Adaptive Shrinkage**
   We introduce a novel adaptation technique, **Adaptive Shrinkage**, which balances population-level trends with individual-specific trajectories. Population and subject-specific models share a common deep kernel that aligns features across subjects in the same latent space, providing a unified feature space for both population and subject-specific dynamics.
   By leveraging this common representation, the Adaptive Shrinkage method enables effective personalization within the deep kernel regression framework for longitudinal biomarkers. This dual focus—capturing global trends while refining individual trajectories—provides a robust foundation for modeling heterogeneous progression patterns and enhances the interpretability and adaptability of the framework to diverse biomarker dynamics.

3. **Robust Generalizability**
   We demonstrate that both the population models and the Adaptive Shrinkage function generalize well to external clinical studies, showcasing their robustness across diverse contexts.
   - **AIBL**: Includes an older cohort primarily composed of AD patients.
   - **OASIS**: Focuses on a younger cohort with a majority of healthy controls.
   - **PreventAD**: Targets a healthier, younger population for early pre-symptomatic detection of AD.
   These differences highlight the adaptability of our framework to varying age groups, disease stages, and clinical goals.

4. **Interpretable Transformation Function**
   The transformation function captures meaningful neuroimaging patterns associated with biomarker progression, offering insights into the underlying processes driving disease dynamics.

5. **Explainable Adaptive Shrinkage**
   Adaptive Shrinkage is inherently explainable (Section 3.5), ensuring transparency in its operation and enhancing trustworthiness for clinical applications where informed decision-making is paramount.

6. **Reproducibility**
   To promote reproducibility and validation, we provide publicly available code and validation on open datasets as ADNI.

*Continues...*

---

---

> ### Author Response · Authors · 2024-12-03
> **Final Remarks**
>
> ---
>
> ## Discussion on the Potential of Adaptive Shrinkage Estimation for Personalized Deep Kernel Regression
>
> ### Introduction to Adaptive Shrinkage
> The **Adaptive Shrinkage** Function introduces a novel and versatile approach to predicting biomarker progression trajectories. By integrating two predictors that share a deep kernel, it projects inputs into a common feature space, achieving feature alignment across subjects. This shared representation informs both population-level trends and individual-specific trajectories, enabling our model to account for inter-subject variability while maintaining consistency with broader population dynamics.
>
> ### Advancements Over Prior Models
> Unlike prior models (e.g., Rudovic et al., 2019; Koval et al., 2021), which focus solely on predicting the next timepoint(s), our method uniquely addresses the entire trajectory from the baseline acquisition. It forecasts future biomarker values while also correcting noisy observations from baseline onward, producing robust and reliable estimates of progression rates (slopes). This dual functionality—integrating prediction and correction—marks a significant advancement over existing progression models. By providing comprehensive and accurate trajectory estimates, our approach enhances clinical decision-making in monitoring and managing neurodegenerative diseases.
>
> ### Clinical Applications
> Robust estimates of progression rates hold potential to impact several areas. They enable early diagnosis by detecting early signs of disease progression and support personalized treatment planning by facilitating tailored interventions. They improve clinical trial design and subject stratification by targeting specific subgroups, and they allow for more precise evaluation of therapeutic efficacy over time. These capabilities make our deep kernel framework with the Adaptive Shrinkage a valuable tool for both researchers and clinicians. (Maheux et al., 2023)
>
> ### Flexibility and Minimal Assumptions
> The Adaptive Shrinkage Function is highly flexible and operates without stringent assumptions about the backbone models or the transformation function $\Phi$. It requires only the predictive means, variances of the population DKGP and ss-DKGP, and the observation times, making it adaptable to diverse setups. This minimal set of requirements broadens its applicability and allows for seamless integration into other deep kernel regression frameworks, such as Stochastic Variational Deep Kernel Learning (SVDKL).
>
> ### Versatility in Diverse Frameworks
> This flexibility underscores the function's ability to integrate additional inductive biases, such as temporal smoothness, hierarchical modeling, or multi-modal data integration, to address more intricate modeling challenges. Its compatibility with diverse setups highlights its potential to enhance the scope and precision of deep kernel regression frameworks.
>
> ### Future Directions
> Future advancements aim to extend this framework by incorporating more expressive transformation functions $\Phi$, such as attention mechanisms, to capture complex relationships in the data. Additionally, moving to stochastic variational deep kernel techniques will help scale its application to more intricate scenarios, further unlocking the potential of Adaptive Shrinkage Estimation in personalized progression modeling. This remains an active area that we plan to explore.
>
> ---
> *Continues...*

---

> ### Author Response · Authors · 2024-12-03
> **Final Remarks**
>
> ---
>
> ## **Summary of Changes and Additions**
>
> ### **Main Manuscript**
>
> #### **Introduction**
> Replaced "monotonic" with "neuroimaging" to better reflect the model's flexibility and clarify that no monotonicity constraints are imposed.
> Expanded the discussion of differences from related methods, including Rudovic et al., 2019.
> Enhanced the contributions section to emphasize the critical role of deep kernel regression models in our approach.
>
> #### **Methods**
> Reformatted by moving some equations inline for improved readability without altering the content.
>
> #### **Results**
>
> **Figures and Tables**
> - Updated **Figure 2** and **Figure 4** with confidence intervals (CIs) for clearer comparisons. Captions were updated accordingly. (enPj)
> - Increased the legend font size in **Figure 3** for better readability. (7rm7)
> - Updated **Table 5** to include CIs in parentheses for clarity and consistency. (enPj)
>
> **Narrative Updates**
> Added a paragraph in the "Generalization to External Clinical Studies" section, describing results with CIs while maintaining our conclusion that the proposed method outperforms all baselines. (enPj)
> Integrated supplementary insights into the main narrative, including:
> - Quantitative comparisons against baselines, stratified errors, and qualitative results (Section 3.3).
> - Error analysis based on the number of observations (Section 3.4). (enPj)
> - Performance comparisons across external clinical studies (Section 3.5). (enPj)
> - Illustrative examples of Adaptive Shrinkage in test samples (Section 3.6). (7rm7, Q2Nf)
>
> ### **Supplementary Material**
>
> We expanded the supplementary material significantly, adding **15 new pages** to address reviewers' concerns comprehensively. The new sections include:
>
> - **Section A**: Extended Related Works in Predictive and Progression Modeling and Clinical Applications. (Q2Nf)
> - **Section E.3**: Statistical significance tests on performance metrics across progression statuses for all baselines for a more comprehensive presentation of the results. (Q2Nf)
> - **Section E.4**: Stratified performance analysis by covariates (Sex, APOE4 Alleles, and Education Years) with barplots showing mean AE and CIs.
> - **Section E.5**: Extended discussion on the limitations of baseline models. (Q2Nf)
> - **Section E.6**: Qualitative examples highlighting how our adaptation algorithm operates, illustrated with predicted trajectories and elaboration on clinical applications. This improves the interpretation of our results and connects them to real clinical scenarios. (Q2Nf, 7rm7)
>
> - **Section E.7**: By expanding the training datasets from 2 to 7 studies (3504 subjects in total), we developed p-DKGP model and Adaptive Shrinkage Function for the SPARE-AD biomarker. We conducted error analysis based on observation counts, presented SPARE-AD trajectory examples, and provided detailed result interpretations. (enPj)
>
> - **Section E.8**: Demonstrated how p-DKGP effectively learns smooth progression curves without requiring imputation, handling missing longitudinal biomarkers internally. Quantitative results confirmed its superiority over Linear Interpolation for interpolation tasks. (Q2Nf)
>
> - **Section E.9**: Latent Space Analysis revealed that the transformation function captures atrophy patterns linked to AD and brain aging, projecting input features into a space optimized for progression prediction.
>
> ---
>
> We sincerely thank the reviewers for their valuable feedback, which has improved the clarity, presentation, and communication of the methodological and clinical significance of our work. During the discussion period, we had the opportunity to engage with **only one** of the four reviewers despite **multiple** follow-up requests for further feedback or responses to our rebuttal. We sincerely hope that this lack of engagement will not adversely affect or bias the decision-making process.
> We hope the updates, along with the expanded supplementary material, meet your expectations and contribute towards a favourable decision on our submission.
>
> Regards,
>
> Authors of Submission 5004

---

### Meta-Review · Area_Chair_myaa · 2024-12-24

**Metareview:**

The proposed work provides a framework for modelling individual patient trajectories (e.g., in disease progression) across longitudinal data, using (deep) kernel regression. Their proposed regularization is adaptively set, and the authors provide a family of derived variants for subject-specific fitting.

Overall, the response from reviewers has been relatively mild but generally positive. Reviewers noted that the paper is well written and generally appears to have good results (modulo language noted by `enPj`), but that the contribution is difficult to define in light of previous GP work and previous disease progression literature.

By scores alone, this is definitionally borderline. However, after discussion, I find that this paper has value to both the methodological community and the disease modelling/longitudinal imaging community. This, combined with a clear description of their own method as well as deep kernel regression and its regularization, leads me to recommend acceptance as a poster.

**Additional Comments On Reviewer Discussion:**

I strongly recommend implementing the edits suggested by `enPj` and answering the questions posed by `7rm7`. I see that some of them have been responded to in the rebuttals and that the manuscript appendix has been extended significantly (15 additional pages), but a few points have not been addressed, even in text.

I also specifically agree with `enPj`'s assessment that "consistent superiority" is incorrect. This is not to say that prospective shrinkage estimates aren't useful, but it is simply factually incorrect based off of the presented results to claim that this method is empirically better than baseline methods in all cases.

---

### Decision · Program_Chairs · 2025-01-22

Accept (Poster)